# Uni-Map: Unified Camera-LiDAR Perception for Robust HD Map Construction

## Abstract

High-definition (HD) map construction methods play a vital role in providing precise and comprehensive static environmental information essential for autonomous driving systems. The primary sensors used are cameras and LiDAR, with input configurations varying among camera-only, LiDAR-only, or camera-LiDAR fusion based on cost-performance considerations, while fusion-based methods typically perform the best. However, current methods face two major issues: high costs due to separate training and deployment for each input configuration, and low robustness when sensors are missing or corrupted. To address these challenges, we propose the Unified Robust HD Map Construction Network (Uni-Map), a single model designed to perform well across all input configurations. Our approach designs a novel Mixture Stack Modality (MSM) training scheme, allowing the map decoder to learn effectively from camera, LiDAR, and fused features. We also introduce a projector module to align Bird's Eye View features from different modalities into a shared space, enhancing representation learning and overall model performance. During inference, our model utilizes a switching modality strategy to adapt seamlessly to any input configuration, ensuring compatibility across various modalities. To evaluate the robustness of HD map construction methods, we designed 13 different sensor corruption scenarios and conducted extensive experiments comparing Uni-Map with state-of-the-art methods. Experimental results show that Uni-Map outperforms previous methods by a significant margin across both normal and corrupted modalities, demonstrating superior performance and robustness. Notably, our unified model surpasses independently trained camera-only, LiDAR-only, and camera-LiDAR MapTR models with a gain of 4.6, 5.6, and 5.6 mAP on the nuScenes dataset, respectively. The source code will be released.

## 1 Introduction

Online high-definition (HD) map provides abundant and precise static environmental information about the driving scenes, which is fundamental for planning and navigation in autonomous driving systems. Cameras and LiDAR are the predominant sensors, offering semantic-rich image data and explicit geometric information from point clouds, respectively. HD map construction models can be categorized into three groups based on input configurations: camera-only Qiao et al. (2023); Ding et al. (2023); Yuan et al. (2024); Liu et al. (2024a); Li et al. (2024), LiDAR-only Li et al. (2022a); Liu et al. (2023a), and camera-LiDAR fusion Liao et al. (2023a;b); Zhou et al. (2024) models. As illustrated in Fig. 1 (a)-(c), HD map construction methods with different input configurations have been widely studied and deployed in real-world systems based on different cost-effective considerations.

However, existing methods entail the training and deployment of separate models for each input configuration, resulting in substantial development, maintenance, and deployment overheads. To address this problem, we propose a novel **Unified Robust HD Map Construction Network (Uni-Map)**, where one trained model can perform well under all input configurations, depicted in Fig. 1(d). Our approach elaborates a novel Mixture Stack Modality (MSM) training scheme during the training phase, allowing the map decoder to glean rich knowledge from the camera, LiDAR, or fused features. Furthermore, we introduce a novel projector module to map Bird's Eye View (BEV) features of different modalities into a shared space. During inference, we present a switching modality strategy enabling precise predictions by Uni-Map when utilizing arbitrary modality inputs. Extensive

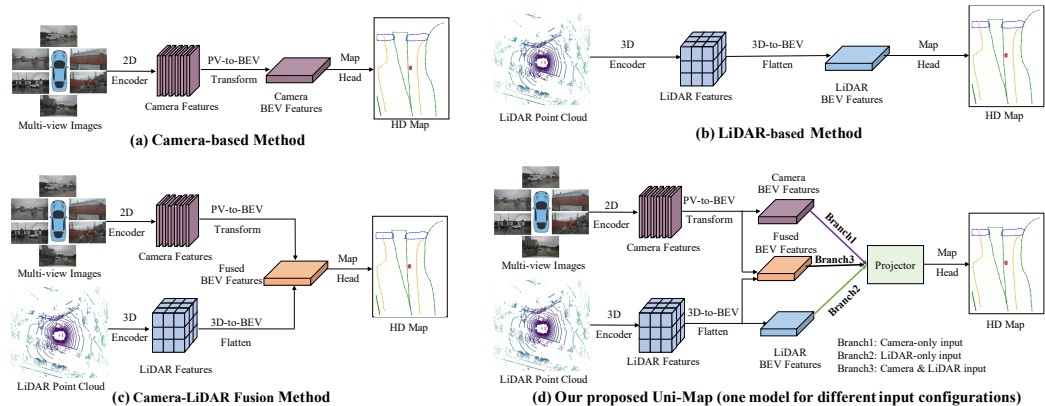

Figure 1: Illustration of the Camera-based method, LiDAR-based method, Camera-LiDAR Fusion method, and the proposed Uni-Map method (one model for different input configurations).

experiments demonstrate that Uni-Map can achieve high performance in different input configurations while reducing the training and deployment costs of the model.

Another critical concern of HD map construction methods for autonomous driving is the model's robustness Kong et al. (2024). While Camera-LiDAR fusion methods have shown promising performance by incorporating information from both modalities Liao et al. (2023a); Zhou et al. (2024); Hao et al. (2024b), existing fusion methods often assume access to complete sensor information, leading to low robustness and potential collapse when sensors are corrupted or missing. To comprehensively evaluate the robustness of the Camera-LiDAR fusion model, we design 13 types of camera-LiDAR corruption combinations that perturb both camera and LiDAR inputs separately or concurrently. These combinations are summarized into 6 cases and illustrated in Fig. 2 (left). We compare Uni-Map with state-of-the-art MapTR Liao et al. (2023a) method, Uni-Map performs more robustly as depicted in Fig. 2 (right), benefiting from the comprehensive feature representations learned by our proposed MSM and aligned by the projector module. Quantitatively, when facing missing camera sensors, Uni-Map still achieves 61.2 mAP, which outperforms the original MapTR Liao et al. (2023a) by +38.7 mAP (61.2 *vs.* 22.5). Experimental results show that Uni-Map exhibits stronger robustness on various multi-sensor corruption types. Importantly, the core components of Uni-Map, *i.e.*, MSM training scheme, projector module, and the switching modality strategy are simple yet effective plug-and-play techniques compatible with existing pipelines.

In summary, the main contributions of this paper are threefold:

- We propose a novel Unified Robust HD Map Construction Network (Uni-Map), which stands out as an all-in-one model to operate on arbitrary input configurations.
- We design a novel Mixture Stack Modality training scheme with a simple yet effective projector module to project the BEV features of different modalities into a shared space, allowing the map decoder to learn strong representation from different modalities and a switching modality strategy to utilize arbitrary modality inputs during inference.
- Our single Uni-Map model beats the state-of-the-art MapTR models independently trained on camera-only, LiDAR-only, and camera-LiDAR fusion modalities with a gain of 4.6, 5.6, and 5.6 mAP, respectively. Moreover, Uni-Map shows much better robustness on 13 types of camera-LiDAR corruption combinations. These benefits extend to various map construction models due to our simple, task-independent designs.

## 2 RELATED WORK

**HD Map Construction.** HD map construction is a prominent and extensively researched area within the field of autonomous driving. According to the input sensor modality, HD map construction models can be categorized into camera-only Liao et al. (2023a); Zhang et al. (2024c); Ding et al. (2023); Liao et al. (2023b); Yuan et al. (2024), LiDAR-only Li et al. (2022a); Liu et al. (2023a) and camera-LiDAR fusion Liao et al. (2023a;b); Zhou et al. (2024); Hao et al. (2024c) models.

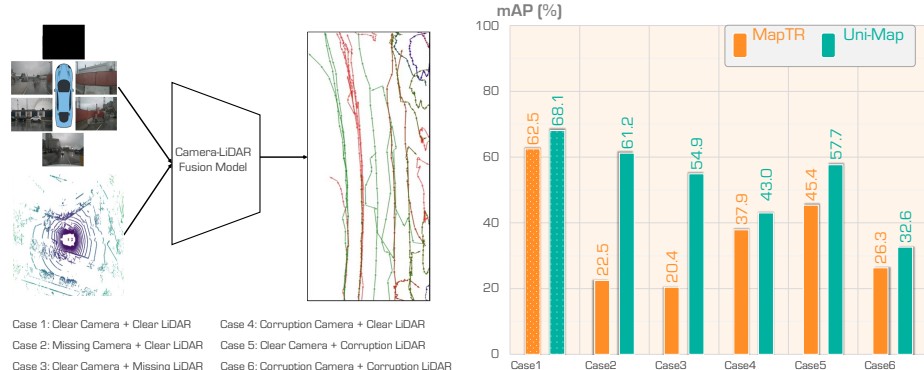

Figure 2: **Uni-Map shows stronger robustness on various multi-sensor corruption types.** We show mAP results for MapTR and Uni-Map models on clean data and each type of multi-sensor corruption. Results show Uni-Map can mitigate the performance drop on sensor missing or corruptions.

Recently, camera-only methods Liu et al. (2024b); Zhang et al. (2024a); Shi et al. (2024); Wang et al. (2024); Zhang et al. (2024b); Peng et al. (2024); Choi et al. (2024); Chen et al. (2024); Jiang et al. (2024); Hu et al. (2024); Hao et al. (2024a) have increasingly employed the Bird's-eye view (BEV) representation as an ideal feature space for multi-view perception due to its remarkable ability to mitigate scale-ambiguity and occlusion challenges. Various techniques have been proposed and utilized to project perspective view (PV) features into the BEV space by leveraging geometric priors, such as LSS Philion & Fidler (2020), Deformable Attention Li et al. (2022b) and GKT Chen et al. (2022). However, camera-only methods suffer from a lack of explicit depth information. LiDAR-only methods Wang et al. (2023); Li et al. (2022a); Liu et al. (2023a); Liao et al. (2023b;a) benefit from the accurate 3D geometric information from the LiDAR input. However, they struggle to deal with data sparsity and sensing noise problems robustly. Recently, camera-LiDAR feature fusion in the unified BEV space has attracted much attention Liao et al. (2023a;b); Zhou et al. (2024); Dong et al. (2024). BEV-level fusion uses two independent streams that encode the raw inputs from the camera and LiDAR sensors into features within the same BEV space. This fusion at the BEV level incorporates complementary modality features, surpassing unimodal input approaches in performance.

While significant progress has been made using various methods with different input configurations (camera-only, LiDAR-only, camera-LiDAR fusion) chosen based on cost-performance considerations, a common challenge persists. Current methods necessitate training and deploying separate models for each input configuration, leading to considerable costs in development, maintenance, and deployment. In this paper, we introduce a novel Unified Robust HD map construction approach to address this issue. This method enables training a single model capable of operating on any input configuration, thereby streamlining the process.

**Robustness Under Sensor Failures.** Sensor failures can significantly impact the accuracy of HD map tasks, thereby jeopardizing the safety of autonomous driving. While Camera-LiDAR fusion methods have shown promising performance, which can make use of both the semantic-rich information from cameras and the explicit geometric information from LiDAR, existing fusion methods often assume access to complete sensor information from both cameras and LiDAR, leading to low robustness in the face of sensor missing or corruptions. This means that their performance may degrade significantly or even fail entirely when sensor data is incomplete or corrupted. Recently, there have been a few studies that focus on benchmarking and improving the robustness under natural corruptions, particularly in various BEV perception algorithms such as 3D object detection Li et al. (2022b); Liu et al. (2023b); Ge et al. (2023), BEV segmentation Zhang et al. (2022); Zhou & Krähenbühl (2022), occupancy prediction Wei et al. (2023b); Huang et al. (2023), and depth estimation Wei et al. (2023a). However, approaches addressing sensor failures for HD map construction are still under exploration.

In this paper, to explore the camera-LiDAR fusion model robustness on the HD map construction task, we design 13 types of camera-LiDAR corruption combinations that perturb both camera and LiDAR inputs separately or concurrently. Our proposed Uni-Map model demonstrates improved robustness under various sensor failure scenarios. To the best of our knowledge, Uni-Map is the first study to explore the robustness of HD map construction task under multi-sensor corruptions.

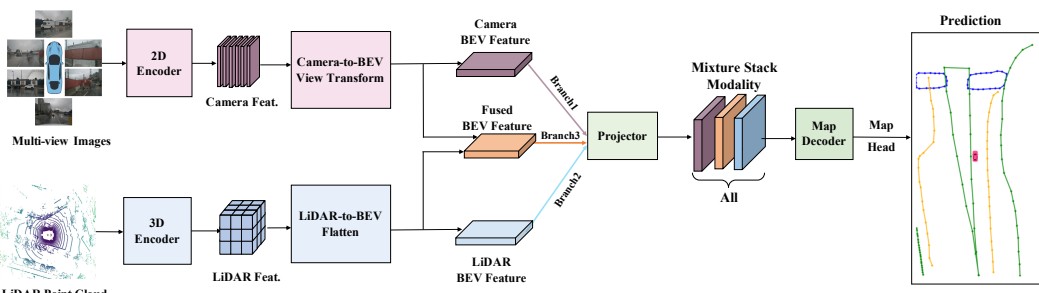

Figure 3: An overview of Uni-Map framework. First, we extract features from multi-modal sensor inputs and convert them into a unified bird's-eye view (BEV) space efficiently using view transformations. Then, we design a novel Mixture Stack Modality (MSM) scheme with a projector module to re-project the BEV features of different modalities into a shared space. Finally, the mixture stack BEV features are fed into a shared decoder and prediction heads for HD Map construction.

## 3 METHODOLOGY

Uni-Map pursues a novel Unified Robust HD Map construction approach, which can train an all-in-one model capable of operating with various input configurations. For this purpose, we feed the model decoder with the features from all input configurations at the training stage and process one specific feature based on the deployed input configuration during inference. The overview framework of Uni-Map is shown in Fig. 3. Given different sensory inputs, we first apply modality-specific encoders to extract their features. These multi-modal features are then transformed into a unified BEV representation that preserves both geometric and semantic information. Then, we incorporate a projector module to align BEV features from different modalities into a shared space, thereby enhancing representation learning. Additionally, we introduce a novel Mixture Stack Modality training scheme, enabling the map decoder module to glean rich knowledge from the camera, LiDAR, or fused features. Specifically, the mixture stack BEV features are fed into the decoder and prediction heads for the HD Map construction task. During inference, we employ a switching modality strategy, enabling Uni-Map to make precise predictions using arbitrary modality inputs.

### 3.1 PRELIMINARIES

For notation clarity, we first introduce some symbols and definitions used throughout this paper. Our goal is to design a novel Unified Robust HD map construction framework taking arbitrary modal sensor data $\chi$ as input and predicting vectorized map elements in BEV space, and the types of the map elements (supported types are road boundary, lane divider, and pedestrian crossing). Formally, assume that we have a set of inputs, $\chi = \{Camera, LiDAR\}$, containing multi-view RGB camera images in perspective view, $Camera \in \mathbb{R}^{B \times N^{cam} \times H^{cam} \times W^{cam} \times 3}$, where $B, N^{cam}, H^{cam}, W^{cam}$ denote batch, number of cameras, image height, and image width, respectively, as well as a LiDAR point cloud, $LiDAR \in \mathbb{R}^{B \times P \times 5}$, with number of points $P$. Each point consists of its 3-dimensional coordinates, reflectivity, and beam index. The detailed architectural designs are described as follows.

### 3.2 MAP ENCODER

We build our Map Encoder based on the state-of-the-art HD map construction method MapTR Liao et al. (2023a), which applies modality-specific encoders to extract their features and transforms multi-modal features into a unified BEV representation that preserves both geometric and semantic information. Note that our approach is compatible with other Map Encoders that can also be employed to generate camera-only, LiDAR-only, and camera-LiDAR fusion BEV features.

**Camera to BEV.** For camera images, we first utilize Resnet50 He et al. (2016b) as the backbone to extract the multi-view features. Then we adopt GKT Chen et al. (2022) as the 2D-to-BEV transformation module to convert the multi-view features into BEV space. The generated BEV features can be denoted as $F_{Camera}^{BEV} \in \mathbb{R}^{B \times H \times W \times C}$, where $H, W, C$ represent the height, width, and the number of channels of BEV features, respectively.

**LiDAR to BEV.** For the LiDAR points, we follow SECOND Yan et al. (2018) in using voxelization and a sparse LiDAR encoder. The LiDAR features are projected to BEV space using a flattening operation as in Liu et al. (2023b), to obtain the unified LiDAR BEV representation $F_{LiDAR}^{BEV} \in \mathbb{R}^{B \times H \times W \times C}$.

**Fused BEV.** We utilize a convolution-based fusion method Liao et al. (2023a); Zhou et al. (2024) to effectively fuse the BEV features from both camera and LiDAR sensors. More specifically, we utilize concatenation followed by convolution to fuse features from multi-modal BEV feature inputs, $F_{Camera}^{BEV} \in \mathbb{R}^{B \times H \times W \times C}$ and $F_{LiDAR}^{BEV} \in \mathbb{R}^{B \times H \times W \times C}$, resulting in the aggregated features $F_{Fused}^{BEV} \in \mathbb{R}^{B \times H \times W \times C}$.

### 3.3 MIXTURE STACK MODALITY AND PROJECTOR

In this section, we first introduce the projector module that aims to align BEV features from different modalities into a shared space, thereby enhancing representation learning and overall model performance. Then, we offer the details of the Mixture Stack Modality (MSM) training scheme, which enables the map decoder module to learn rich knowledge from the camera, LiDAR, or fused features.

**Projector Module.** After input sensor features converted to the shared BEV representation, we can easily obtain the BEV features of the three modalities, *i.e.*, $F_{Camera}^{BEV} \in \mathbb{R}^{B \times H \times W \times C}$, $F_{LiDAR}^{BEV} \in \mathbb{R}^{B \times H \times W \times C}$ and $F_{Fused}^{BEV} \in \mathbb{R}^{B \times H \times W \times C}$. While in the same space, camera BEV features, LiDAR BEV features, and fused BEV features can still be misaligned to some extent due to the inaccurate depth in the view transformer and the large modality gap (See Fig. 7 (a)). Existing works Liang et al. (2022); Liu et al. (2023b) show the phenomenon of modal gaps, *i.e.*, the features of different BEV modalities usually focus on completely separate regions in BEV space. Thus, we propose a projector module to align BEV features from different modalities into a shared space (see the *Remarks* below), thereby enhancing representation learning. To address this issue, we project BEV features of different modalities into a new shared space via a learnable projector $projector(\cdot)$, *i.e.*,

$$\hat{F}_{camera}^{BEV} = projector(F_{camera}^{BEV}), \tag{1}$$

$$\hat{F}_{LiDAR}^{BEV} = projector(F_{LiDAR}^{BEV}), \tag{2}$$

$$\hat{F}_{Fused}^{BEV} = projector(F_{Fused}^{BEV}), \tag{3}$$

where $projector(\cdot)$ is the multi-layer linear perceptron (MLP) function. Note that, the BEV features of different modalities use a shared projector, and the details are discussed in the ablation experiments.

**Mixture Stack Modality Training Scheme.** The map decoder module in existing HD map construction methods is typically trained using BEV features from a single mode, limiting it to one input configuration. To address this limitation and ensure that a single trained model can perform well across all input configurations, we introduce a novel Mixture Stack Modality training scheme after the projector module. Specifically, it can be formulated as:

$$\hat{F}_{Stack}^{BEV} = Stack(\hat{F}_{camera}^{BEV}, \hat{F}_{LiDAR}^{BEV}, \hat{F}_{Fused}^{BEV}). \tag{4}$$

Using the MSM scheme, we obtain the stacked multi-modal BEV feature $\hat{F}_{Stack}^{BEV} \in \mathbb{R}^{3B \times H \times W \times C}$, which serves as input for the HD map construction task. Notably, the stacking operation preserves the feature map shape as $H \times W \times C$ by stacking along the batch dimension. This design choice enables seamless integration with the subsequent Map Decoder module in existing methods, such as MapTR Liao et al. (2023a). Consequently, our method operates in a plug-and-play manner, ensuring easy implementation and compatibility.

*Remarks*: The MSM scheme offers three key advantages. First, by stacking BEV features from different modalities that share the *same* map decoder and ground truth labels, the projector module is supervised (via gradient back-propagation) to implicitly align BEV features from different modalities in the shared feature space. Second, inputting stacked BEV features into the same map decoder increases the diversity of the BEV feature space accessible to the decoder module, thereby improving the model's generalization ability and robustness across different input configurations. Third, this scheme allows the map decoder module to process BEV features of different modalities. As a result, Uni-Map can flexibly handle various input configurations during inference.

Figure 4: Illustration of the switching modality strategy.

## 3.4 FULL OBJECTIVE AND INFERENCE

**Overall Training.** We follow the MapTR Liao et al. (2023a) model's training loss function, which is composed of three parts, including the classification loss $\mathcal{L}_{cls}$, the point2point loss $\mathcal{L}_{p2p}$, and the edge direction loss $\mathcal{L}_{dir}$. Combining these loss terms, the overall objective function can be formulated as:

$$\mathcal{L} = \lambda_1 \mathcal{L}_{cls} + \lambda_2 \mathcal{L}_{p2p} + \lambda_3 \mathcal{L}_{dir}, \tag{5}$$

where $\lambda_1$, $\lambda_2$ and $\lambda_3$ are hyper parameters for balancing these terms. For all experiments, $\lambda_1$ is set to 2, $\lambda_2$ is set to 5, and $\lambda_3$ is set to $5e^{-3}$.

**Inference Phase.** During inference, our model utilizes a switching modality strategy to seamlessly adapt to arbitrary modality inputs, ensuring compatibility across various input configurations. The switching modality strategy can be formulated as:

$$\hat{F}_{\text{Select}}^{\text{BEV}} = \begin{cases} \hat{F}_{camera}^{BEV}, & \text{if } Camera \ only \ sensor \ input, \\ \hat{F}_{lidar}^{BEV}, & \text{if } LiDAR \ only \ sensor \ input, \\ \hat{F}_{fused}^{BEV}, & \text{if } Camera \ and \ LiDAR \ are \ both \ obtained. \end{cases} \tag{6}$$

This switching strategy simulates real-world scenarios where sensors may be missing during the inference phase. As shown in Fig. 4, if the LiDAR data is not available due to uninstallation or damage, we select camera BEV feature $\hat{F}_{camera}^{BEV}$ as the input of the map decoder, and vice versa. Moreover, if Camera and LiDAR data are both obtained, we select Fused BEV features $\hat{F}_{fused}^{BEV}$ as the input for the map decoder. As a result, Uni-Map is compatible with any of the above three input configurations, thereby enhancing its practicality in autonomous driving.

## 4 EXPERIMENTS

### 4.1 EXPERIMENTAL SETTINGS

**Datasets.** We evaluate our method on the widely-used challenging nuScenes Caesar et al. (2020) dataset following the standard setting of previous methods Liao et al. (2023a); Zhou et al. (2024). The nuScenes dataset contains 1,000 sequences of recordings collected by autonomous driving cars. Each sample is annotated at 2Hz and contains 6 camera images covering $360°$ horizontal FOV of the ego-vehicle. Following Liao et al. (2023a); Li et al. (2022a); Gao et al. (2024), three kinds of map elements are chosen for fair evaluation – pedestrian crossing, lane divider, and road boundary.

**Evaluation Metrics.** We adopt the evaluation metrics consistent with previous works Liao et al. (2023a); Li et al. (2022a); Zhang et al. (2024c), where average precision (AP) is used to evaluate the map construction quality and Chamfer distance $D_{Chamfer}$ determines the matching between predictions and ground truth. We calculate the $AP_\tau$ under several $D_{Chamfer}$ thresholds ($\tau \in T = \{0.5m, 1.0m, 1.5m\}$), and then average across all thresholds as the final mean AP (*mAP*) metric. The perception ranges are $[-15.0m, 15.0m]/[-30.0m, 30.0m]$ for X/Y-axes.

**Implementation Details.** Uni-Map is trained with 4 NVIDIA RTX A6000 GPUs. During the training phase, the GT labels are duplicated twice and stacked to form $3B$ batch dimension, matching with the stacked feature map from Eq. 4. The design choice of the MSM scheme is discussed in the ablation studies. For the projector module, we use a two-layer perceptron whose dimension is C->C/2->C. We adopt the AdamW optimizer Loshchilov & Hutter (2019) for all our experiments. We set the mini-batch size to 16, and use a step-decayed learning rate with an initial value of $4e^{-3}$. The inference time is measured on a single NVIDIA RTX A6000 GPU with batch size 1.

Table 1: **Comparisons with state-of-the-art methods on nuScenes val set.** "L" and "C" represent LiDAR and camera, respectively. "Effi-B0", "R50", "PP", and "Sec" are short for EfficientNet-B0 Tan & Le (2019), ResNet50 He et al. (2016a), PointPillars Lang et al. (2019) and SECOND Yan et al. (2018), respectively. Note that Uni-Map (MapModel) means our method is integrated into an existing MapModel. Best viewed in color. nfan

| Method | Modality | BEV Encoder | Backbone | Epoch | $AP_{ped.}$ | $AP_{div.}$ | $AP_{bou.}$ | mAP ↑ |
|---|---|---|---|---|---|---|---|---|
| HDMapNet Li et al. (2022a) | C | NVT | Effi-B0 | 30 | 14.4 | 21.7 | 33.0 | 23.0 |
| VectorMapNet Liu et al. (2023a) | C | IPM | R50 | 110 | 36.1 | 47.3 | 39.3 | 40.9 |
| PivotNet Ding et al. (2023) | C | PersFormer | R50 | 30 | 53.8 | 58.8 | 59.6 | 57.4 |
| BeMapNet Qiao et al. (2023) | C | IPM-PE | R50 | 30 | 57.7 | 62.3 | 59.4 | 59.8 |
| MapVR Zhang et al. (2024c) | C | GKT | R50 | 24 | 47.7 | 54.4 | 51.4 | 51.2 |
| MapTRv2 Liao et al. (2023b) | C | BEVPoolv2 | R50 | 24 | 59.8 | 62.4 | 62.4 | 61.5 |
| StreamMapNet Yuan et al. (2024) | C | BEVFormer | R50 | 30 | 61.7 | 66.3 | 62.1 | 63.4 |
| MapTR Liao et al. (2023a) | C | GKT | R50 | 24 | 46.3 | 51.5 | 53.1 | 50.3 |
| HIMap Zhou et al. (2024) | C | BEVFormer | R50 | 24 | 62.2 | 66.5 | 67.9 | 65.5 |
| **Uni-Map (MapTR)** | **C** | **GKT** | **R50** | **24** | **52.1** | **57.5** | **55.2** | **54.9** |
| **Uni-Map (HIMap)** | **C** | **BEVFormer** | **R50** | **24** | **64.5** | **68.2** | **68.3** | **67.0** |
| VectorMapNet Liu et al. (2023a) | L | - | PP | 110 | 25.7 | 37.6 | 38.6 | 34.0 |
| MapTRv2 Liao et al. (2023b) | L | - | Sec | 24 | 56.6 | 58.1 | 69.8 | 61.5 |
| MapTR Liao et al. (2023a) | L | - | Sec | 24 | 48.5 | 53.7 | 64.7 | 55.6 |
| HIMap Zhou et al. (2024) | L | - | Sec | 24 | 54.8 | 64.7 | 73.5 | 64.3 |
| **Uni-Map (MapTR)** | **L** | **-** | **Sec** | **24** | **56.5** | **57.8** | **69.4** | **61.2** |
| **Uni-Map (HIMap)** | **L** | **-** | **Sec** | **24** | **65.3** | **69.5** | **77.8** | **70.8** |
| MapTRv2 Liao et al. (2023b) | C & L | BEVPoolv2 | R50 & Sec | 24 | 65.6 | 66.5 | 74.8 | 69.0 |
| MapTR Liao et al. (2023a) | C & L | GKT | R50 & Sec | 24 | 55.9 | 62.3 | 69.3 | 62.5 |
| HIMap Zhou et al. (2024) | C & L | BEVFormer | R50 & Sec | 24 | 71.0 | 72.4 | 79.4 | 74.3 |
| **Uni-Map (MapTR)** | **C & L** | **GKT** | **R50 & Sec** | **24** | **64.4** | **66.8** | **73.2** | **68.1** |
| **Uni-Map (HIMap)** | **C & L** | **BEVFormer** | **R50 & Sec** | **24** | **73.6** | **75.3** | **81.2** | **76.7** |

Table 2: Comparison of MapTR Liao et al. (2023a) and Uni-Map in terms of accuracy, model size, training epochs and training time on nuScenes dataset. Note that only one Uni-Map model is trained while three MapTR models (MapTR-C, MapTR-L, and MapTR-F) are trained for different input configurations. † represents using the total time of training three MapTR models to train our Uni-Map model.

| Method | Camera-only (mAP) | LiDAR-only (mAP) | Camera & LiDAR (mAP) | Params(MB) | Epoch | Training Time |
|---|---|---|---|---|---|---|
| MapTR-C | 50.3 | — | — | 35.9 | 24 | 13h55m |
| MapTR-L | — | 55.6 | — | 14.3 | 24 | 9h7m |
| MapTR-F | — | — | 62.5 | 39.8 | 24 | 15h44m |
| **Uni-Map (MapTR)** | **54.9** | **61.2** | **68.1** | **39.9** | **24** | **21h57m** |
| **Uni-Map (MapTR)†** | **57.2** | **64.5** | **70.4** | **39.9** | **42** | **38h44m** |

## 4.2 COMPARISON WITH THE STATE-OF-THE-ARTS

With the same settings, we compare our method with several state-of-the-art methods across three categories, *i.e.*, camera-only methods, LiDAR-only methods, and camera-LiDAR fusion methods. Specifically, we integrate our Uni-Map into two recent methods, MapTR Liao et al. (2023a) and HIMap Zhou et al. (2024), where we insert the projector module into these models and apply the MSM training scheme. Moreover, to fairly evaluate the effectiveness, we train the same epochs as the original model. It's noteworthy that while three MapTR/HIMap models need to be trained for different input configurations, our Uni-Map model only requires training once. As shown in Tab. 1, our Uni-Map significantly improves the performance compared to the original models. Specifically, Uni-Map (MapTR) outperforms independently trained camera-only, LiDAR-only, and camera-LiDAR MapTR models on NuScenes with a large gain of 4.6, 5.6, and 5.6 mAP, under the respective input configurations, respectively. Based on the previous state-of-the-art HIMap, our all-in-one model surpasses HIMap-C, HIMap-L, and HIMap-F by 1.5, 6.5, and 2.4 mAP respectively, establishing a new state-of-the-art in vectorized map reconstruction. Results for more datasets like Argoverse2 Wilson et al. (2021) are shown in the supplementary material A.3. All these results prove the effectiveness of our design.

**Model Size, Training Time, GPU Memory and Inference Speed.** To systematically evaluate the effectiveness of our proposed Uni-Map model, we comprehensively analyze it in terms of accuracy, model size, training time, and inference speed. The experimental results are shown in Tab. 2 and Appendix Tab. 6-Tab. 7. The experimental results reveal some interesting findings: (1) Compared with MapTR, Uni-Map performs much better in all input configurations in both single-class APs

Table 3: Ablation study on the MSM training scheme. The mAP values on nuScenes val set are reported. 'Mean' represents the average mAP of three input configurations.

| Random Select | Mixture Stack | Projector | Camera-only | LiDAR-only | Camera & LiDAR | Mean |
|:---:|:---:|:---:|:---:|:---:|:---:|:---:|
| ✗ | ✗ | ✗ | 20.4 | 22.5 | 62.5 | 35.1 |
| ✓ | ✗ | ✗ | 36.9 | 47.5 | 62.9 | 49.1 |
| ✗ | ✓ | ✗ | **53.7** | **59.4** | **67.9** | **60.3** |
| ✓ | ✗ | ✓ | 45.6 | 55.3 | 61.2 | 54.0 |
| ✗ | ✓ | ✓ | **54.9** | **61.2** | **68.1** | **61.4** |

Table 4: Ablation study on Projector Module. The mAP values on nuScenes val set are reported. 'Mean' represents the average mAP of three input configurations.

| Method | Camera-only | LiDAR-only | Camera&LiDAR | Mean |
|:---|:---:|:---:|:---:|:---:|
| Baseline (w/o projector) | 53.7 | 59.4 | 67.9 | 60.3 |
| Variant 1: Independent Projector | 53.6 | **62.2** | 67.6 | 61.1 |
| Variant 2:Partially Shared Projector | 53.3 | 61.5 | 68.0 | 60.9 |
| Variant 3: Skip Shared Projector | 53.4 | 61.7 | 68.0 | 61.0 |
| **Variant 4: Shared Projector (Ours)** | **54.9** | 61.2 | **68.1** | **61.4** |

and the overall mAP. Note that only one Uni-Map model is trained while three MapTR models (MapTR-C, MapTR-L, and MapTR-F) are trained for different input configurations. Thus, we use the same computational budget of training three MapTR models to train our Uni-Map model, and the resulting Uni-Map model (last row of Tab. 2) beats independently trained camera-only, LiDAR-only, and camera-LiDAR fusion MapTR models with a larger gain of 6.9, 8.9, 7.9 mAP, under the respective input configurations. (2) In terms of model size, our Uni-Map model only increases the number of parameters by 0.1MB compared to the MapTR-F model, as shown in Tab. 2. It is more parameter-efficient than deploying the three models simultaneously in practice. (3) In terms of GPU Memory and inference speed, the quantities of our Uni-Map and MapTR are almost the same, as shown in Appendix Tab. 6-Tab. 7. All in all, the Uni-Map model achieves significant performance improvements over the strong MapTR baseline with less training time and fewer parameters (for various input configurations), while maintaining the same inference speed and memory footprint.

## 4.3 ABLATION STUDIES

**Analysis of the MSM training scheme.** To systematically evaluate the effectiveness of the MSM training scheme, we train the model using different schemes and report the mAP results in Tab. 3. In addition to MSM, we also introduce the Random Select Modality (RSM) training scheme that receives inputs from one BEV feature map randomly selected among $\hat{F}_{camera}^{BEV}, \hat{F}_{LiDAR}^{BEV}, \hat{F}_{Fused}^{BEV}$. In the main ablation study, we design the following model variants: (1) We train the model without the projector module and any of the RSM and MSM training schemes. (2) We train the model without the projector module using RSM or MSM training schemes, respectively. (3) We train the model with the projector module using RSM or MSM training schemes, respectively. The experimental results reveal some interesting findings: (1) The results of both RSM and MSM schemes are significantly better than the Baseline model (only learned/seen the BEV features of one modality), verifying the effectiveness of learning with rich knowledge from different BEV features to improve the generalization ability of the map decoder. (2) The results of the RSM training scheme are inferior to the MSM training scheme under both settings (with and without the Projector). This demonstrates the MSM training scheme's advantage in enhancing the map decoder's effective use of camera, LiDAR, and fused features. This increases the diversity of the BEV feature space, resulting in a high-performance integrated model.

**Analysis on projector module.** We investigate the design choice of the projector module in our method. The ablation variants include Independent Projector, Partially Shared Projector, Skip Shared Projector, and Shared Projector (the default setting). The detailed formulation of the variant projector module is in the supplementary material A.1. As shown in Tab. 4, the experimental results reveal some interesting findings: (1) Using different projector variants consistently outperforms the baseline model, implying that using the simple projector module can facilitate learning better feature representations. This can be owing to the fact that our model uses the same map decoder and ground truth labels to promote feature alignment in this latent space. (2) Using a shared projector module consistently outperforms other projector variants. It is reasonable that using BEV feature information

from different modalities to perform gradient updates on a shared projector, rather than on multiple projectors, aligns BEV features from different modalities more effectively. These observations validate the effectiveness of the projector module in aligning BEV features from different modalities into a shared space, thereby enhancing representation learning and overall model performance.

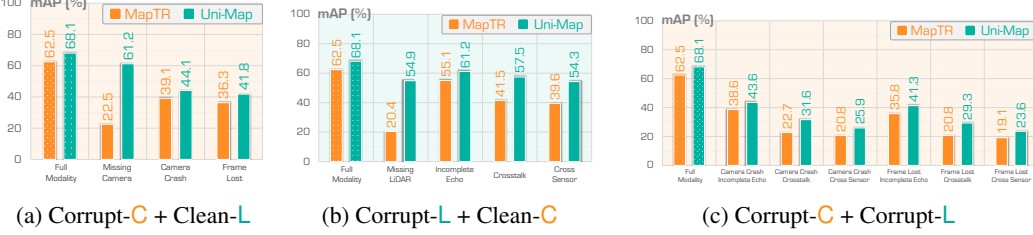

(a) Corrupt-C + Clean-L      (b) Corrupt-L + Clean-C      (c) Corrupt-C + Corrupt-L

Figure 5: The result of multi-sensor corruption on MapTR *vs.* Uni-Map (MapTR) fusion model.

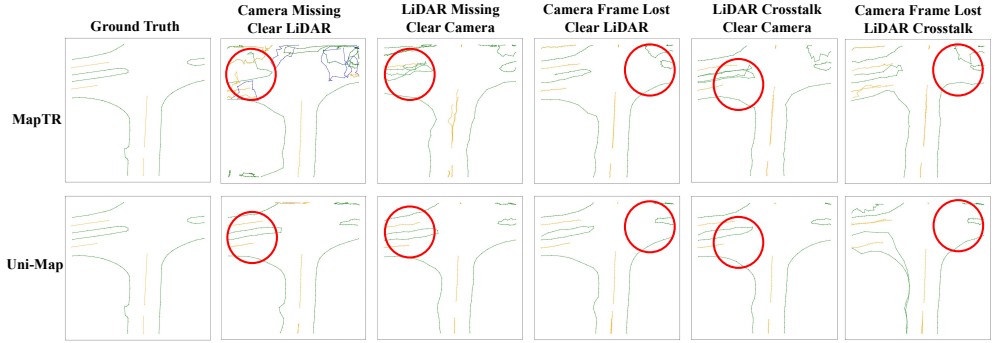

Figure 6: Qualitative results of the nuScenes val set on the MapTR and UniMap models respectively.

### 4.4 ROBUSTNESS OF MULTI-SENSOR CORRUPTIONS

To explore the camera-LiDAR fusion model robustness, we design 13 types of camera-LiDAR corruption combinations that perturb both camera and LiDAR inputs separately or concurrently. Camera-LiDAR corruption combinations are grouped into camera-only corruptions, LiDAR-only corruptions, and their combinations, covering the majority of real-world corruption cases. The definition of multi-sensor corruption is detailed in A.2. Fig. 5 shows the results of three Camera-LiDAR corruption combinations. We have the following observations. (1) In the sensor missing , Uni-Map can prevent the model from collapsing owing to the switching modality strategy. Quantitatively, when facing a missing LiDAR sensor, Uni-Map still achieves 54.9 mAP, which outperforms the original MapTR Liao et al. (2023a) by 34.5 mAP. (2) In case of the corruption of the camera and LiDAR sensor individually or simultaneously, Uni-Map shows stronger robustness. For example, in the face of camera frame lost and LiDAR crosstalk, compared to the MapTR fused model, the Uni-Map model achieved significant improvements in 8.5 mAP (29.3 *vs.* 20.8). These results demonstrate that the MSM training scheme enhances the generalization ability of the map decoder. By stacking BEV features from different modalities into the same map decoder, the diversity of the BEV feature space accessible to the decoder increases, thereby improving the model's robustness. All in all, Uni-Map shows stronger robustness on our designed 13 types of camera-LiDAR corruption combinations.

### 4.5 VISUALIZATION

**Qualitative Results.** To further analyze the effectiveness of our Uni-Map model, we compare it with MapTR Liao et al. (2023a) and present the qualitative results in Fig. 6. We compare the predicted vectorized HD map results of different settings, including the camera sensor missing, LiDAR sensor missing, camera frame lost and clear LiDAR, LiDAR crosstalk and clear camera, and camera frame lost with LiDAR crosstalk. We observe that the baseline MapTR predictions are highly erroneous,

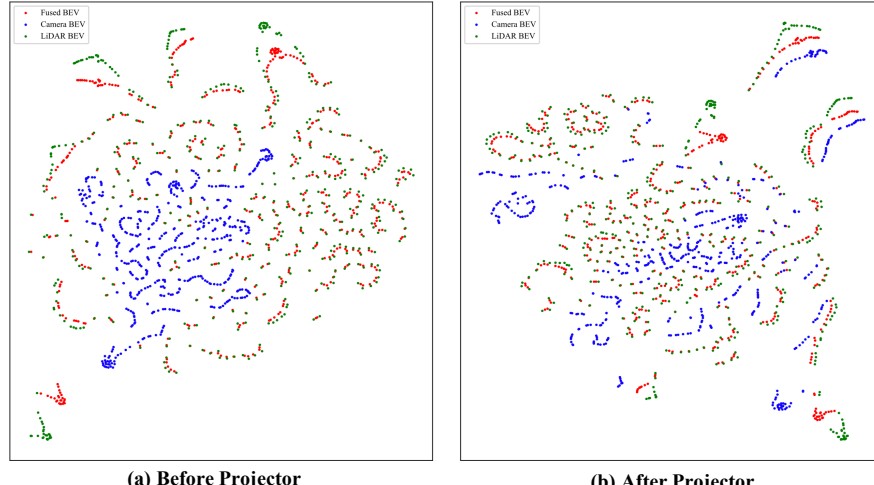

(a) Before Projector                   (b) After Projector

Figure 7: The t-SNE visualizations of (a) Before Projector module and (b) After Projector module. Red/Blue/green denotes fused BEV feature/camera BEV feature/LiDAR BEV feature. After the projector module, the BEV features from different modalities are aligned in a shared space, *e.g*, red, blue, and green circles are close together after the projector module (best viewed in color).

whereas our Uni-Map model can already correct significant errors in the baseline predictions in all settings. All in all, our model shows significant advantages in clear and various corruption situations.

**t-SNE.** We randomly choose 500 samples on the nuScenes dataset and show the tSNE Van der Maaten & Hinton (2008) visualizations of (a) Before Projector module and (b) After Projector module in Fig. 7. Red/Blue/green denotes fused BEV feature/camera BEV feature/LiDAR BEV feature. As can be seen, Fig. 7 (a) Before Projector module shows that blue and red/green features are clearly separated, indicating that although in the same space, camera BEV features, LiDAR BEV features, and fused BEV features can still be misaligned to some extent due to the inaccurate depth in the view transformer and the large modality gap. Fig. 7 (b) After the projector module, the BEV features from different modalities are aligned in a shared space, *i.e.*, red, blue, and green circles are close together after the projector module.

## 5 CONCLUSION

In this paper, we propose a novel Unified Robust HD Map Construction Network (Uni-Map), which can train an all-in-one model to operate on arbitrary input configurations. The core components of Uni-Map, *i.e.* MSM training scheme, projector module, and the switching modality strategy, are simple yet effective plug-and-play techniques compatible with existing pipelines. Extensive experiments demonstrate that Uni-Map can achieve high performance in different input configurations while reducing the training and deployment costs of the model. Moreover, Uni-Map shows stronger robustness on our designed 13 types of camera-LiDAR corruption combinations. We hope that our method can be applied to more autonomous driving perception tasks.

**Ethics Statement.** Our work can boost the performance and robustness of HD map construction task. Although our method significantly improves the robustness of the HD map model, the overall robustness is still low. Special caution is needed in deploying our methods onto vehicles on the road to ensure safety. Therefore, future research is necessary to further investigate more advanced robustness methods.

**Reproducibility.** To ensure the reproducibility of our work, we have included a comprehensive Reproducibility Statement. Specifically, for the novel model and algorithms presented in this work, we will make them open source upon paper acceptance. Additionally, all multi-sensor corruption details and more experimental results can be found in Appendix A. For the datasets used in our experiments, we follow the standard protocol of the open source work MapTR Liao et al. (2023a) . This Reproducibility Statement is intended to guide readers to the relevant resources that will aid in replicating our work, ensuring transparency and clarity throughout.

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

# A  APPENDIX / SUPPLEMENTAL MATERIAL

This supplementary material provides additional details on the proposed method and experimental results that could not be included in the main manuscript due to page limitations.

- Section A.1 discusses details of different variant projector modules.
- Section A.2 provides additional details of the multi-sensor corruptions.
- Section A.3 complements Argoverse2 dataset experiment results and corresponding analysis.
- Section A.4 presents the results of the switching modality strategy on the original MapTR fusion model.
- Section A.5 offers more experimental results regarding model robustness.
- Section A.6 offers 3D object detection results to prove the generalization ability of the Uni-Map.
- Section A.7 includes more visualization results to prove the effectiveness of the Uni-Map.

## A.1  VARIANT PROJECTOR MODULE

After input sensor features converted to the shared BEV representation, we can easily obtain the BEV features of the three modalities, *i.e.*, $F_{Camera}^{BEV} \in \mathbb{R}^{B \times H \times W \times C}$, $F_{LiDAR}^{BEV} \in \mathbb{R}^{B \times H \times W \times C}$ and $F_{Fused}^{BEV} \in \mathbb{R}^{B \times H \times W \times C}$. While in the same space, camera BEV features, LiDAR BEV features, and fused BEV features can still be misaligned to some extent due to the inaccurate depth in the view transformer and the large modality gap (See Fig. 7 (a) ). Existing works Liang et al. (2022); Liu et al. (2023b) show the phenomenon of modal gaps, *i.e.*, the features of different BEV modalities usually focus on completely separate regions in BEV space. Thus, we propose a projector module to align BEV features from different modalities into a shared space, thereby enhancing representation learning. To address this issue, we project BEV features of different modalities into a new shared space via a learnable projector $projector(\cdot)$.

**Shared Projector.** The Shared Projector formula can be written as:

$$\hat{F}_{camera}^{BEV} = projector(F_{camera}^{BEV}), \tag{7}$$

$$\hat{F}_{LiDAR}^{BEV} = projector(F_{LiDAR}^{BEV}), \tag{8}$$

$$\hat{F}_{Fused}^{BEV} = projector(F_{Fused}^{BEV}), \tag{9}$$

where $projector(\cdot)$ is the two-layer linear perceptron function. Note that, the BEV features of different modalities use a shared projector module.

**Partially Shared Projector.** The main difference from the shared projector is that the first linear layer of the partially shared projector learns three modes independently, and the second linear layer is shared.

**Independent Projector.** The Independent Projector formula can be written:

$$\hat{F}_{camera}^{BEV} = projector_1(F_{camera}^{BEV}), \tag{10}$$

$$\hat{F}_{LiDAR}^{BEV} = projector_2(F_{LiDAR}^{BEV}), \tag{11}$$

$$\hat{F}_{Fused}^{BEV} = projector_3(F_{Fused}^{BEV}), \tag{12}$$

where $projector(\cdot)$ is the multi-layer linear perceptron function. Note that, the BEV features of different modalities use different projector modules.

**Skip Shared Projector.** The Skip Shared Projector formula can be written as:

$$\hat{F}_{camera}^{BEV} = projector(F_{camera}^{BEV}) + F_{camera}^{BEV}, \tag{13}$$

$$\hat{F}_{LiDAR}^{BEV} = projector(F_{LiDAR}^{BEV}) + F_{LiDAR}^{BEV}, \tag{14}$$

$$\hat{F}_{Fused}^{BEV} = projector(F_{Fused}^{BEV}) + F_{Fused}^{BEV}, \tag{15}$$

where $projector(\cdot)$ is the two-layer linear perceptron function. Note that, the BEV features of different modalities use a shared skip projector module.

Table 5: Description and severity level setups in camera/LiDAR corruption simulations. Camera Crash (Camera), Frame Lost (Frame), Crosstalk, Incomplete Echo (Echo), and Cross-Sensor (Sensor).

| Corruption | Description | Parameter | Easy | Moderate | Hard |
|---|---|---|---|---|---|
| Camera | droping view images | number of dropped camera | 2 | 4 | 5 |
| Frame | droping temporal frames | probability of frame dropping | 2/6 | 4/6 | 5/6 |
| Crosstalk | light impluses interference | percentage | 0.03 | 0.07 | 0.12 |
| Echo | imcomplete LiDAR readings | drop ratio | 0.75 | 0.85 | 0.95 |
| Sensor | cross sensor data | beam number to drop | 8 | 16 | 20 |

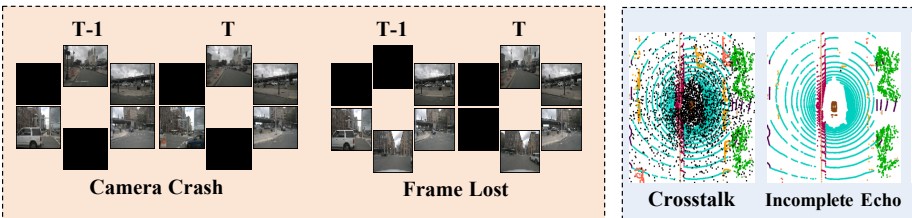

**Camera Corruption**      **LiDAR Corruption**

Figure 8: Visualization results of camera/LiDAR sensor corruptions.

## A.2 MULTI-SENSOR CORRUPTIONS

To explore the camera-LiDAR fusion model robustness, we design 13 types of camera-LiDAR corruption combinations that perturb both camera and LiDAR inputs separately or concurrently. Camera-LiDAR corruption combinations are grouped into camera-only corruptions, LiDAR-only corruptions, and their combinations, covering the majority of real-world corruption cases. Specifically, we design 3 types of camera-only corruptions by utilizing the clean LiDAR point data and three camera failure cases such as Unavailable Camera (*all pixel values are set to zero for all RGB images*), Camera Crash, and Frame Lost. Moreover, we design 4 types for LiDAR-only corruptions by utilizing the clean camera data and the corrupted LiDAR data as the input. The LiDAR corruption types include complete LiDAR failure which means LiDAR data are unavailable (*Since no model can work when all points are absent, we approximate this scenario by only retaining a single point as input*), LiDAR Incomplete Echo, LiDAR Crosstalk, and LiDAR Cross-Sensor. Note that our implementation of complete LiDAR failure is close to the real-world situation. Lastly, we design 6 types of camera-LiDAR corruption combinations that perturb both sensor inputs concurrently, using the previously mentioned image/LiDAR sensor failure types. We establish several corruption severity levels (*i.e.*, three levels including easy, moderate, and hard) for each type of corruption. Furthermore, for a comprehensive evaluation, we report metrics for each corruption type by averaging over three severity levels. Description and severity level setups in 2 types of camera corruption and 3 types of LiDAR corruption are shown in Tab. 5. Visualization results of camera/LiDAR sensor corruptions are shown in Fig. 8.

## A.3 RESULTS ON ARGOVERSE2 DATASET

There are 1000 logs in the Argoverse2 dataset Wilson et al. (2021). Each log contains 15s of 20Hz RGB images from 7 cameras, 10Hz LiDAR sweeps, and a 3D vectorized map. The train, validation, and test sets contain 700, 150, and 150 logs, respectively. Following previous works Liao et al. (2023a); Zhou et al. (2024), we report results on its validation set and focus on the same three map categories as the nuScenes dataset.

Tab. 8 and Tab. 9 show the overall performance of Uni-Map and all the baselines on the Argoverse2 dataset. Compared with MapTR, Uni-Map outperforms all input configurations in both single-class APs and the overall mAP by a significant margin on the Argoverse2 dataset. Note that only one Uni-Map model is trained while three MapTR models (MapTR-C, MapTR-L, and MapTR-F) are trained for different input configurations. Thus, we use the total time of training three MapTR models to train our Uni-Map model, and the resulting Uni-Map model (last row of Tab. 9) beats independently trained camera-only, LiDAR-only, and camera-LiDAR fusion MapTR models with gains of 5.0, 4.8, 6.6 mAP, under the respective input configurations. In a nutshell, Uni-Map shows significant

Table 6: Comparison of MapTR Liao et al. (2023a) and Uni-Map in terms of inference speed (Frames-per-Second).

| Method | Camera-only | LiDAR-only | Camera & LiDAR |
|---|---|---|---|
| MapTR-C | 21.4 | — | — |
| MapTR-L | — | 8.7 | — |
| MapTR-F | — | — | 6.4 |
| Uni-Map (MapTR) | 21.4 | 8.7 | 6.4 |

Table 7: Comparison of MapTR Liao et al. (2023a) and Uni-Map in terms of GPU memory (MB) footprint.

| Method | Camera-only | LiDAR-only | Camera & LiDAR |
|---|---|---|---|
| MapTR-C | 2544 | — | — |
| MapTR-L | — | 9963 | — |
| MapTR-F | — | — | 10607 |
| Uni-Map (MapTR) | 2544 | 9963 | 10607 |

superiority over other baseline methods on the nuScenes and the Argoverse2 datasets, indicating the benefit of our method.

Table 8: **Comparisons with state-of-the-art methods on Argoverse2 dataset.** Note that Uni-Map (MapModel) means our method is integrated into an existing MapModel.

| Method | Modality | BEV Encoder | Backbone | Epoch | $AP_{ped.}$ | $AP_{div.}$ | $AP_{bou.}$ | mAP ↑ |
|---|---|---|---|---|---|---|---|---|
| HDMapNet Li et al. (2022a) | C | NVT | Effi-B0 | 6 | 13.1 | 5.7 | 37.6 | 18.8 |
| VectorMapNet Liu et al. (2023a) | C | IPM | R50 | 24 | 38.3 | 36.1 | 39.2 | 37.9 |
| MapTRv2 Liao et al. (2023b) | C | BEVPoolv2 | R50 | 6 | 62.9 | 72.1 | 67.1 | 67.4 |
| HIMap Zhou et al. (2024) | C | BEVFormer | R50 | 6 | 69.0 | 69.5 | 70.3 | 72.7 |
| MapTR Liao et al. (2023a) | C | GKT | R50 | 6 | 57.9 | 56.9 | 59.2 | 58.0 |
| MapTR Liao et al. (2023a) | L | - | R50 | 6 | 56.1 | 56.7 | 74.9 | 62.5 |
| MapTR Liao et al. (2023a) | C & L | GKT | R50 & Sec | 6 | 65.1 | 61.6 | 75.1 | 67.3 |
| **Uni-Map (MapTR)** | **C** | **GKT** | **R50** | **6** | **60.2** | **62.9** | **62.9** | **62.0** |
| **Uni-Map (MapTR)** | **L** | **-** | **R50** | **6** | **60.0** | **60.0** | **77.8** | **66.0** |
| **Uni-Map (MapTR)** | **C & L** | **GKT** | **R50 & Sec** | **6** | **70.1** | **69.4** | **80.5** | **73.3** |

Table 9: Comparison of MapTR Liao et al. (2023a) and Uni-Map in terms of accuracy, model size, training epochs and training time on the Argoverse2 dataset. Note that only one Uni-Map model is trained while three MapTR models (MapTR-C, MapTR-L, and MapTR-F) are trained for different input configurations. † represents using the total time of training three MapTR models to train our Uni-Map model.

| Method | Camera-only (mAP) | LiDAR-only (mAP) | Camera & LiDAR (mAP) | Params(MB) | Epoch | Training Time |
|---|---|---|---|---|---|---|
| MapTR-C | 58.0 | — | — | 35.9 | 6 | 11h46m |
| MapTR-L | — | 62.5 | — | 14.3 | 6 | 7h38m |
| MapTR-F | — | — | 67.3 | 39.8 | 6 | 13h22m |
| **Uni-Map (MapTR)** | **62.0** | **66.0** | **73.3** | **39.9** | **6** | **18h31m** |
| **Uni-Map (MapTR)†** | **63.0** | **67.2** | **73.9** | **39.9** | **10** | **30h51m** |

Table 10: Results of switching modality strategy on MapTR Fusion model.

| Method | Camera-only (mAP) | LiDAR-only (mAP) | Camera & LiDAR (mAP) |
|---|---|---|---|
| MapTR | 0 | 0 | 62.5 |

## A.4 RESULTS OF MSM ON ORIGINAL MAPTR FUSION MODEL

We use our proposed switching modality strategy on the original MapTR Fusion model on nuScenes dataset. The experimental results are shown in Tab. 10. We are surprised to find that directly using our switching modality strategy in the existing MapTR fusion model, the performance of the camera-only branch and LiDAR branch are zero. Experimental results prove that without using our Mixture Stack Modality (MSM) training scheme and projector module, the model is unable to handle various input configurations. The above experimental phenomena verify the effectiveness and rationality of our MSM training scheme and projector module design.

## A.5 MORE EXPERIMENTAL RESULTS REGARDING MODEL ROBUSTNESS

To explore the camera-LiDAR fusion model robustness, we design 13 types of camera-LiDAR corruption combinations that perturb both camera and LiDAR inputs separately or concurrently. Camera-LiDAR corruption combinations are grouped into camera-only corruptions, LiDAR-only

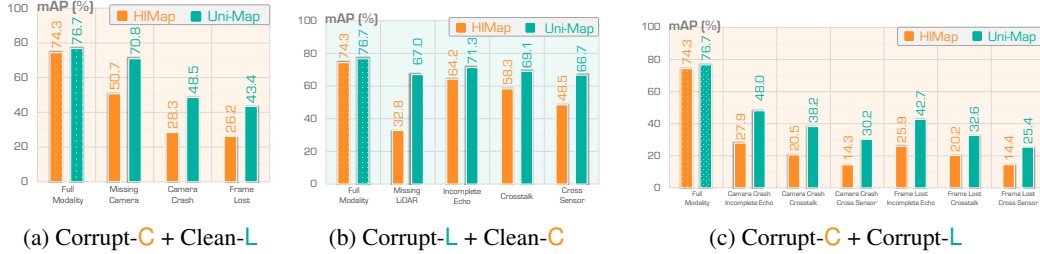

(a) Corrupt-C + Clean-L     (b) Corrupt-L + Clean-C     (c) Corrupt-C + Corrupt-L

Figure 9: The result of multi-sensor corruption on HIMap *vs.* Uni-Map (HIMap) fusion model.

corruptions, and their combinations, covering the majority of real-world corruption cases. Fig. 9 shows the results of three Camera-LiDAR corruption combinations on HIMap Zhou et al. (2024) fusion model. We can find that: (1) In the sensor missing scenario, Uni-Map can still keep the model from collapsing based on our switching modality strategy. Quantitatively, when facing the camera sensor missing case, Uni-Map still achieves 70.8 mAP, which outperforms the original HIMap Zhou et al. (2024) by +20.1 mAP. (2) In case of corruption of camera and LiDAR sensor individually or simultaneously, Uni-Map still shows stronger robustness. For example, in the face of camera crash and LiDAR crosstalk, compared to the MapTR fused model, the Uni-Map model achieved significant improvements in 17.7 mAP (38.2 *vs.* 20.5). All in all, Uni-Map shows stronger robustness on our designed 13 types of camera-LiDAR corruption combinations. Experimental results for all corruption types for MapTR and Uni-Map (MapTR) are shown in Tab. 12-Tab. 14. And, experimental results for all corruption types for HIMap and Uni-Map (HIMap) are shown in Tab. 15-Tab. 17.

Table 11: Comparison of BEVFusion Liu et al. (2023b) and Uni-Map in terms of accuracy on the nuScenes dataset. Note that only one Uni-Map model is trained while three BEVFusion models (BEVFusion-C, BEVFusion-L and BEVFusion-F) are trained for different input configurations.

| Method | Camera-only (mAP/NDS) | LiDAR-only (mAP/NDS) | Camera & LiDAR (mAP/NDS) |
|---|---|---|---|
| BEVFusion-C | 35.6/41.2 | — | — |
| BEVFusion-L | — | 64.7/69.3 | — |
| BEVFusion-F | — | — | 68.5/71.4 |
| **Uni-Map (BEVFusion)** | **39.2/46.1** | **67.3/71.6** | **71.1/73.5** |

## A.6   GENERALIZATION TO 3D OBJECT DETECTION TASK

In order to verify the universality of the Uni-Map method, we thereby generalize our method to the 3D object detection task, to further show its effectiveness on other perception tasks. We select the popular 3D object detection method BEVFusion Liu et al. (2023b) as the baseline model. As shown in the Tab. 11, our Uni-Map consistently improves the performance, compared to the original model. For example, our Uni-Map beats independently trained camera-only, LiDAR-only, and camera-LiDAR fusion models with gains of 3.6/4.9, 2.6/2.3, 2.6/2.1 mAP/NDS, under the respective input configurations. Obviously, our method can be directly utilized in the 3D objection detection task, demonstrating the generalization ability of our method.

## A.7   MORE VISUALIZATION RESULTS

We provide more visualization results of qualitative results. Visualization results of qualitative results are shown in Fig. 10. We observe that in the case of multi-sensor corruption, the source MapTR model predictions are highly incorrect. However, our Uni-Map model can already correct significant errors in the baseline predictions in all settings. Qualitative results demonstrate the superiority of the UniMap model in various corruption scenarios.

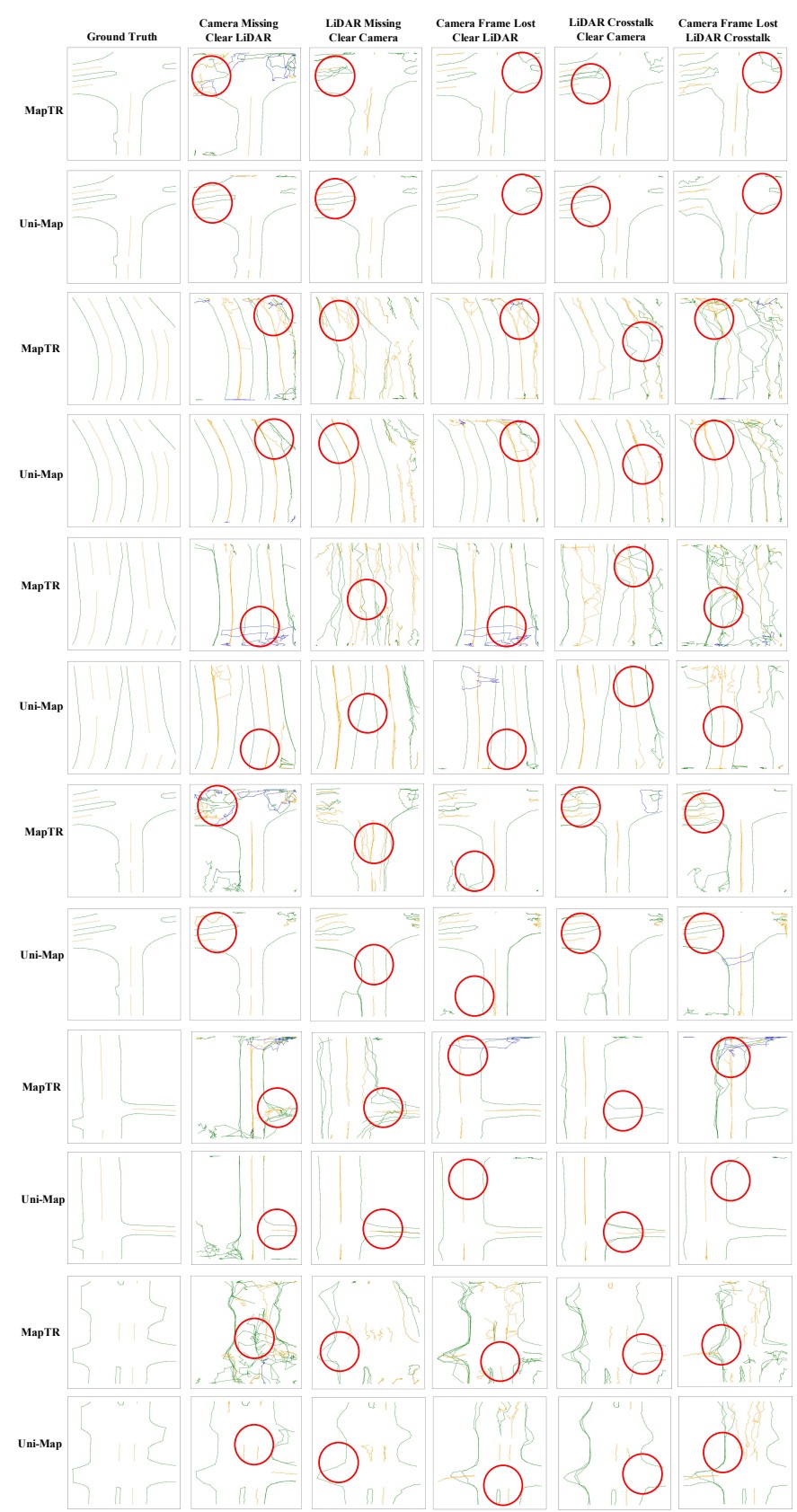

Figure 10: Qualitative results on nuScenes val set.

Table 12: The result of camera-only corruptions on MapTR *vs* Uni-Map (MapTR) fusion model.

| Method | Modality | Camera | LiDAR | $AP_{ped.}$ | $AP_{div.}$ | $AP_{bou.}$ | mAP↑ |
|---|---|---|---|---|---|---|---|
| MapTR Liao et al. (2023a) | C & L | ✔ | ✔ | 55.9 | 62.3 | 69.3 | 62.5 |
| MapTR Liao et al. (2023a) | C & L | ✗ | ✔ | 15.0 | 18.2 | 34.4 | $22.5_{-40.0}$ |
| MapTR Liao et al. (2023a) | C & L | Camera Crash | ✔ | 32.5 | 36.5 | 48.4 | $39.1_{-23.4}$ |
| MapTR Liao et al. (2023a) | C & L | Frame Lost | ✔ | 29.1 | 33.7 | 46.1 | $36.3_{-26.2}$ |
| **Uni-Map (MapTR)** | C & L | ✔ | ✔ | **64.4** | **66.8** | **73.2** | **68.1** |
| **Uni-Map (MapTR)** | C & L | ✗ | ✔ | **56.5** | **57.8** | **69.4** | $\mathbf{61.2}_{-6.9}$ |
| **Uni-Map (MapTR)** | C & L | Camera Crash | ✔ | **40.3** | **40.3** | **51.5** | $\mathbf{44.1}_{-24.0}$ |
| **Uni-Map (MapTR)** | C & L | Frame Lost | ✔ | **37.0** | **38.6** | **49.9** | $\mathbf{41.8}_{-26.3}$ |

Table 13: The result of LiDAR-only corruptions on MapTR *vs* Uni-Map (MapTR) fusion model.

| Method | Modality | Camera | LiDAR | $AP_{ped.}$ | $AP_{div.}$ | $AP_{bou.}$ | mAP↑ |
|---|---|---|---|---|---|---|---|
| MapTR Liao et al. (2023a) | C & L | ✔ | ✔ | 55.9 | 62.3 | 69.3 | 62.5 |
| MapTR Liao et al. (2023a) | C & L | ✔ | ✗ | 20.7 | 27.4 | 13.1 | $20.4_{-42.1}$ |
| MapTR Liao et al. (2023a) | C & L | ✔ | Incomplete Echo | 47.9 | 55.2 | 62.2 | $55.1_{-7.4}$ |
| MapTR Liao et al. (2023a) | C & L | ✔ | Crosstalk | 36.7 | 42.5 | 45.3 | $41.5_{-21.0}$ |
| MapTR Liao et al. (2023a) | C & L | ✔ | Cross-Sensor | 33.9 | 42.9 | 42.0 | $39.6_{-22.9}$ |
| **Uni-Map (MapTR)** | C & L | ✔ | ✔ | **64.4** | **66.8** | **73.2** | **68.1** |
| **Uni-Map (MapTR)** | C & L | ✔ | ✗ | **52.1** | **57.5** | **55.2** | $\mathbf{54.9}_{-13.2}$ |
| **Uni-Map (MapTR)** | C & L | ✔ | Incomplete Echo | **56.5** | **61.3** | **65.9** | $\mathbf{61.2}_{-6.9}$ |
| **Uni-Map (MapTR)** | C & L | ✔ | Crosstalk | **53.3** | **58.2** | **60.9** | $\mathbf{57.5}_{-10.6}$ |
| **Uni-Map (MapTR)** | C & L | ✔ | Cross-Sensor | **50.5** | **55.4** | **57.2** | $\mathbf{54.3}_{-13.8}$ |

Table 14: The result of camera and LiDAR corruptions on MapTR *vs* Uni-Map (MapTR) fusion model.

| Method | Modality | Camera | LiDAR | $AP_{ped.}$ | $AP_{div.}$ | $AP_{bou.}$ | mAP↑ |
|---|---|---|---|---|---|---|---|
| MapTR Liao et al. (2023a) | C & L | ✔ | ✔ | 55.9 | 62.3 | 69.3 | 62.5 |
| MapTR Liao et al. (2023a) | C & L | Camera Crash | Incomplete Echo | 32.4 | 35.6 | 47.8 | $38.6_{-23.9}$ |
| MapTR Liao et al. (2023a) | C & L | Camera Crash | Crosstalk | 19.7 | 21.6 | 26.9 | $22.7_{-39.8}$ |
| MapTR Liao et al. (2023a) | C & L | Camera Crash | Cross-Sensor | 18.4 | 20.8 | 23.2 | $20.8_{-41.7}$ |
| MapTR Liao et al. (2023a) | C & L | Frame Lost | Incomplete Echo | 28.9 | 32.8 | 45.5 | $35.8_{-26.7}$ |
| MapTR Liao et al. (2023a) | C & L | Frame Lost | Crosstalk | 16.9 | 19.9 | 25.5 | $20.8_{-41.7}$ |
| MapTR Liao et al. (2023a) | C & L | Frame Lost | Cross-Sensor | 15.8 | 19.4 | 22.2 | $19.1_{-43.4}$ |
| **Uni-Map (MapTR)** | C & L | ✔ | ✔ | **64.4** | **66.8** | **73.2** | **68.1** |
| **Uni-Map (MapTR)** | C & L | Camera Crash | Incomplete Echo | **40.3** | **39.7** | **50.8** | $\mathbf{43.6}_{-24.5}$ |
| **Uni-Map (MapTR)** | C & L | Camera Crash | Crosstalk | **29.8** | **28.7** | **36.4** | $\mathbf{31.6}_{-36.5}$ |
| **Uni-Map (MapTR)** | C & L | Camera Crash | Cross-Sensor | **24.5** | **24.6** | **28.8** | $\mathbf{25.9}_{-42.2}$ |
| **Uni-Map (MapTR)** | C & L | Frame Lost | Incomplete Echo | **36.9** | **37.8** | **49.2** | $\mathbf{41.3}_{-26.8}$ |
| **Uni-Map (MapTR)** | C & L | Frame Lost | Crosstalk | **26.3** | **27.3** | **34.3** | $\mathbf{29.3}_{-38.8}$ |
| **Uni-Map (MapTR)** | C & L | Frame Lost | Cross-Sensor | **20.9** | **23.3** | **26.6** | $\mathbf{23.6}_{-44.5}$ |

Table 15: The result of camera-only corruptions on HIMap *vs* Uni-Map (HIMap) fusion model.

| Method | Modality | Camera | LiDAR | $AP_{ped.}$ | $AP_{div.}$ | $AP_{bou.}$ | mAP ↑ |
|---|---|---|---|---|---|---|---|
| HIMap Zhou et al. (2024) | C & L | ✔ | ✔ | 71.0 | 72.4 | 79.4 | 74.3 |
| HIMap Zhou et al. (2024) | C & L | ✗ | ✔ | 40.9 | 46.4 | 74.7 | $50.7_{-23.6}$ |
| HIMap Zhou et al. (2024) | C & L | Camera Crash | ✔ | 36.3 | 27.7 | 20.9 | $28.3_{-46.0}$ |
| HIMap Zhou et al. (2024) | C & L | Frame Lost | ✔ | 29.9 | 25.0 | 23.8 | $26.2_{-48.1}$ |
| **Uni-Map (HIMap)** | C & L | ✔ | ✔ | **73.6** | **75.3** | **81.2** | **76.7** |
| **Uni-Map (HIMap)** | C & L | ✗ | ✔ | **65.3** | **69.5** | **77.8** | $\mathbf{70.8}_{-5.9}$ |
| **Uni-Map (HIMap)** | C & L | Camera Crash | ✔ | **42.5** | **47.6** | **55.5** | $\mathbf{48.5}_{-28.2}$ |
| **Uni-Map (HIMap)** | C & L | Frame Lost | ✔ | **36.7** | **42.3** | **51.1** | $\mathbf{43.4}_{-33.3}$ |

Table 16: The result of LiDAR-only corruptions on HIMap *vs* Uni-Map (HIMap) fusion model.

| Method | Modality | Camera | LiDAR | $AP_{ped.}$ | $AP_{div.}$ | $AP_{bou.}$ | mAP ↑ |
|---|---|---|---|---|---|---|---|
| HIMap Zhou et al. (2024) | C & L | ✔ | ✔ | 71.0 | 72.4 | 79.4 | 74.3 |
| HIMap Zhou et al. (2024) | C & L | ✔ | ✗ | 30.7 | 38.7 | 29.0 | $32.8_{-41.5}$ |
| HIMap Zhou et al. (2024) | C & L | ✔ | Incomplete Echo | 59.1 | 63.7 | 69.9 | $64.2_{-10.1}$ |
| HIMap Zhou et al. (2024) | C & L | ✔ | Crosstalk | 54.1 | 57.5 | 63.4 | $58.3_{-16.0}$ |
| HIMap Zhou et al. (2024) | C & L | ✔ | Cross-Sensor | 44.2 | 50.7 | 50.8 | $48.5_{-25.8}$ |
| **Uni-Map (HIMap)** | C & L | ✔ | ✔ | **73.6** | **75.3** | **81.2** | **76.7** |
| **Uni-Map (HIMap)** | C & L | ✔ | ✗ | **64.5** | **68.2** | **68.3** | $\mathbf{67.0}_{-9.7}$ |
| **Uni-Map (HIMap)** | C & L | ✔ | Incomplete Echo | **68.0** | **70.8** | **75.0** | $\mathbf{71.3}_{-5.4}$ |
| **Uni-Map (HIMap)** | C & L | ✔ | Crosstalk | **65.9** | **68.9** | **72.6** | $\mathbf{69.1}_{-7.6}$ |
| **Uni-Map (HIMap)** | C & L | ✔ | Cross-Sensor | **63.8** | **67.4** | **69.1** | $\mathbf{66.7}_{-10}$ |

Table 17: The result of camera and LiDAR corruptions on HIMap *vs* Uni-Map (HIMap) fusion model.

| Method | Modality | Camera | LiDAR | $AP_{ped.}$ | $AP_{div.}$ | $AP_{bou.}$ | mAP ↑ |
|---|---|---|---|---|---|---|---|
| HIMap Zhou et al. (2024) | C & L | ✔ | ✔ | 71.0 | 72.4 | 79.4 | 74.3 |
| HIMap Zhou et al. (2024) | C & L | Camera Crash | Incomplete Echo | 36.2 | 26.9 | 20.5 | $27.9_{-46.4}$ |
| HIMap Zhou et al. (2024) | C & L | Camera Crash | Crosstalk | 29.2 | 19.3 | 12.9 | $20.5_{-53.8}$ |
| HIMap Zhou et al. (2024) | C & L | Camera Crash | Cross-Sensor | 23.1 | 13.8 | 5.9 | $14.3_{-60.0}$ |
| HIMap Zhou et al. (2024) | C & L | Frame Lost | Incomplete Echo | 29.9 | 24.4 | 23.5 | $25.9_{-48.4}$ |
| HIMap Zhou et al. (2024) | C & L | Frame Lost | Crosstalk | 23.6 | 18.9 | 18.0 | $20.2_{-54.1}$ |
| HIMap Zhou et al. (2024) | C & L | Frame Lost | Cross-Sensor | 17.7 | 14.3 | 11.2 | $14.4_{-59.9}$ |
| **Uni-Map (HIMap)** | C & L | ✔ | ✔ | **73.6** | **75.3** | **81.2** | **76.7** |
| **Uni-Map (HIMap)** | C & L | Camera Crash | Incomplete Echo | **42.4** | **46.7** | **54.8** | $\mathbf{48.0}_{-28.7}$ |
| **Uni-Map (HIMap)** | C & L | Camera Crash | Crosstalk | **35.1** | **36.6** | **42.8** | $\mathbf{38.2}_{-38.5}$ |
| **Uni-Map (HIMap)** | C & L | Camera Crash | Cross-Sensor | **28.9** | **30.8** | **31.0** | $\mathbf{30.2}_{-46.5}$ |
| **Uni-Map (HIMap)** | C & L | Frame Lost | Incomplete Echo | **36.6** | **41.2** | **50.3** | $\mathbf{42.7}_{-34.0}$ |
| **Uni-Map (HIMap)** | C & L | Frame Lost | Crosstalk | **29.2** | **31.3** | **37.5** | $\mathbf{32.6}_{-44.1}$ |
| **Uni-Map (HIMap)** | C & L | Frame Lost | Cross-Sensor | **23.9** | **25.9** | **26.4** | $\mathbf{25.4}_{-51.3}$ |

