# OpenReview forum: "Uni-Map: Unified Camera-LiDAR Perception for Robust HD Map Construction"
_ICLR.cc/2025/Conference — Submitted to ICLR 2025_

### Official Review · Reviewer_js2w · 2024-10-31

**Soundness:** 3
**Presentation:** 3
**Contribution:** 3
**Rating:** 6
**Confidence:** 4

**Summary:**

Summary:

They proposed a novel Unified Robust HD Map Construction Network (UniMap):

1. A novel Mixture Stack Modality (MSM) training scheme is proposed to allow the map decoder to glean rich knowledge from the camera, LiDAR, or fused features.

2. A novel projector module is presented to map Bird’s Eye View (BEV) features of different modalities into a shared space.

3. A switching modality strategy is designed to enable precise predictions by Uni-Map when utilizing arbitrary modality inputs.

4. Extensive experiments demonstrate that Uni-Map can achieve high performance in different input configurations while reducing the training and deployment costs of the model.

**Strengths:**

Strengths:

1. Very careful and detailed analysis of HD map construction task.
2. Very clear figure presentation (i.e. Fig. 2 and Fig. 3 and Fig. 6). The readers are easy to follow the manuscript.
3. Very extensive experiments are conducted in Experiment section and Appendix.
4. Problem configuration is meaningful to the field of autonomous driving.

These strengths reflect that the authors are the experts in this field.

**Weaknesses:**

Weaknesses:

1. Method does not have sufficient theoretical analysis, or theoretical insight. Method section contains lots of engineering details.
2. Why the switching modality strategy can be seamlessly adapted to arbitrary modality inputs?
3. Why MLPs in Eqs. (1-3) work well?

Although extensive expierments show the effectiveness of the proposed method, there lack sufficient theoretical analysis or convincing explanations in their method section.

**Questions:**

Although extensive expierments show the effectiveness of the proposed method, there lack sufficient theoretical analysis or convincing explanations in their method section. Based on this fact, I have to rate this paper as marginally above the acceptance threshold.

---

> ### Author Response · Authors · 2024-11-19
> **Thanks and Response to Reviewer js2w**
>
> Thank you for reviewing our paper and for your positive feedback, especially your comment describing our method as "**_meaningful to the field of autonomous driving and novel_**." We will integrate all your suggestions in the revised version of the paper.
>
> >Q1: Theoretical analysis and insight regarding the method.
>
> **A1:**  Thank you for your valuable feedback regarding the theoretical analysis. We appreciate your insights and would like to clarify our insights further:
>
> **Insight of the Projector Module:**  While in the same space, camera BEV features, LiDAR BEV features, and fused BEV features can still be misaligned to some extent due to the inaccurate depth in the view transformer and the large modality gap (See Fig.7 (a)).  To address this, we propose a projector module to align BEV features from different modalities into a shared space thereby enhancing representation learning. As shown in Fig.7, before the Shared Projector module, the camera BEV and LiDAR/Fused BEV features are clearly separated, however after the shared projector module, the BEV features from different modalities are well aligned in a shared/similar space. Therefore, designing a projector module to align BEV features from different modalities into a shared space is essential for enhancing the overall performance of our model.
>
> **Insight of the MSM Scheme:**   First, by stacking BEV features from different modalities that share the same map decoder and ground truth labels, the projector module is supervised (via gradient back-propagation) to implicitly align BEV features from different modalities in the shared feature space. Second, inputting stacked BEV features into the same map decoder increases the diversity of the BEV feature space accessible to the decoder module, thereby improving the model’s generalization ability and robustness across different input configurations. Third, this scheme allows the map decoder module to process BEV features of different modalities. As a result, Uni-Map can flexibly handle various input configurations during inference.
>
> We hope this explanation clarifies the theoretical insights behind our method and addresses your concerns.
>
> >Q2: Why the switching modality strategy can be seamlessly adapted to arbitrary modality inputs?
>
> **A2:** Thanks for your comments. The Uni-Map model trained with camera and lidar data is evaluated with different input configurations (camera/lidar/camera & lidar). Specifically, during inference, our model utilizes a switching modality strategy to seamlessly adapt to arbitrary modality inputs, ensuring compatibility across various input configurations. The switching modality strategy can be formulated as Eq. 6. This switching strategy simulates real-world scenarios where sensors may be missing during the inference phase. As illustrated in Fig. 4, the map decoder can adapt to various input scenarios: camera BEV features are used when LiDAR data is unavailable due to removal or damage, LiDAR features are used when camera data is absent, and fused BEV features are employed when both data sources are available. This flexibility enables the decoder to handle individual modality features or their fusion seamlessly. As a result, Uni-Map supports all three input configurations, enhancing its practicality for autonomous driving. Notably, this modality-switching strategy does not affect inference speed or memory usage. Experimental settings will be detailed in the revised version.
>
> >Q3: Why MLPs in Eqs. (1-3) work well?
>
> A3: Thanks for your comments. First, by stacking BEV features from different modalities that **_share_** the same map decoder and ground truth labels, the projector module is **_supervised_** (via gradient back-propagation) to **_implicitly align_** BEV features from different modalities in the **_shared feature space_**. After alignment, we concatenate the features along the batch, and inputting the stacked BEV features into the same map decoder enhances the diversity of the BEV feature space, improving the model’s generalization and robustness across various input configurations. As can be seen, Fig.7(a) Before Projector module shows that camera BEV (blue) and LiDAR/Fusion BEV (red/green) features are clearly separated, indicating that although in the same space, camera BEV features, LiDAR BEV features, and fused BEV features can still be misaligned to some extent due to the inaccurate depth in the view transformer and the large modality gap. Fig.7 (b) After the projector module, the BEV features from different modalities are well aligned in a shared/similar space, i.e., red (Fused BEV), blue (Camera BEV), and green (LiDAR) circles are close together after the projector module. Moreover, Tab.3 shows that we can significantly improve the model performance by training the model with the projector module. Therefore, it is demonstrated that the MLP-based projector work well and can align BEV features from different modalities into a shared space.

---

> > ### Author Response · Authors · 2024-11-27
> > **Looking forward to further feedback**
> >
> > Dear Reviewer  js2w,
> >
> > Thank you again for your valuable comments and suggestions, which are very helpful to us. We have responded to the proposed concerns. We hope that it has helped to address the concerns you have raised in your review.
> >
> > We understand that this is quite a busy period, so we sincerely appreciate it if you could take some time to return further feedback on whether our responses resolve your concerns. If you have any further questions, we are more than happy to address them before the conclusion of the rebuttal phase.
> >
> > Best,
> >
> > The Authors

---

> > > ### Comment · Reviewer_js2w · 2024-11-30
> > > **Feedback to Authors**
> > >
> > > Thank you very much for the response. I would rate Accept after checking your response.

---

> > > > ### Author Response · Authors · 2024-11-30
> > > > **Thanks and Response to Reviewer js2w**
> > > >
> > > > Thank you very much for your valuable suggestions and positive feedback! We will revise the final version according to these constructive discussions.

---

### Official Review · Reviewer_scTZ · 2024-11-03

**Soundness:** 4
**Presentation:** 3
**Contribution:** 3
**Rating:** 8
**Confidence:** 4

**Summary:**

This paper introduces a new method for HD map construction. The proposed approach employs a simple but effective scheme (such as Projector Module and MSM Training) to project the BEV features of different modalities into a shared space, which leads to notable improvement for map construction under camera-only, Lidar-only and camera-LiDAR fusion settings. Besides, this approach also shows robustness on various multi-sensor corruption types.

**Strengths:**

1. This paper is well-written and easy to follow.
2. This paper provides extensive qualitative and quantitative experiments to support their claims.
3. This method is motivated and effective, which could also benefit other information fusion research, such as RGB-D segmentation.

**Weaknesses:**

1. From the Table 1 in  Experiment section,  previous state-of-the-art method for HD map construction is HIMap, which is different from the statement in Line 32 and Line 96.

**Questions:**

1. Is it appropriate to put Figure 9 from the Appendix to the main body of the paper and put Table 4 to the Appendix?

---

> ### Author Response · Authors · 2024-11-19
> **Thanks and Response to Reviewer scTZ**
>
> Thank you for reviewing our paper and for the positive feedback, particularly your comment that our method is ‘**_motivated and effective_**’. We will incorporate all your suggestions into the revised version of the paper.
>
> >Q1: From the Table 1 in Experiment section, previous state-of-the-art method for HD map construction is HIMap, which is different from the statement in Line 32 and Line 96.
>
> **A1**: Thank you for your insightful feedback. In the current version of our paper, lines 32 and 96 present experimental results related to the popular HD map construction method-MapTR. To clarify this in the revised version, we will make the following modifications:
>
> **Line 32 will be revised to**: Notably, our unified model surpasses independently trained camera-only, LiDAR-only, and camera-LiDAR state-of-the-art HIMap models with gains of 1.5, 6.5, and 2.4 mAP on the nuScenes dataset, respectively.
>
> **Line 96 will be revised to:** Our single Uni-Map model outperforms the state-of-the-art HIMap models that are independently trained on camera-only, LiDAR-only, and camera-LiDAR fusion modalities, achieving gains of 1.5, 6.5, and 2.4 mAP, respectively.
> We appreciate your suggestion and believe these changes will enhance the clarity of our findings.
>
> >Q2: Is it appropriate to put Figure 9 from the Appendix to the main body of the paper and put Table 4 to the Appendix?
>
> **A2**: Thank you for your suggestion regarding the placement of Figure 9 and Table 4. We agree that moving Figure 9 to the main body of the paper could enhance its visibility and relevance to the discussion. Conversely, we will relocate Table 4 to the Appendix, as it contains supplementary information that may not be essential to the main narrative. We appreciate your input and will make these adjustments in the revised version.

---

> > ### Comment · Reviewer_scTZ · 2024-11-26
> >
> > Thank you for your response and corrections. I believe that this paper not only advances the field of HD map construction but also significantly contributes to other multimodal prediction tasks. Therefore, I maintain my positive assessment and recommend acceptance.

---

> > > ### Author Response · Authors · 2024-11-26
> > > **Thanks and Response to Reviewer scTZ**
> > >
> > > Thank you very much for your valuable suggestions and positive feedback! We will revise the final version according to these constructive discussions.

---

### Official Review · Reviewer_dUuw · 2024-11-04

**Soundness:** 2
**Presentation:** 3
**Contribution:** 2
**Rating:** 6
**Confidence:** 4

**Summary:**

This study addresses the challenges of generalizability and robustness in HD map construction by proposing Uni-Map, a model capable of handling different input configurations, including camera-only (C), LiDAR-only (L), and fused camera-LiDAR (C+L) modalities. This model is based on the MapTR model and employs a Mixture Stack Modality (MSM) training scheme, wherein each sample in the training batch is augmented to include all input configurations (C/L/C+L) by stacking along the batch dimension. The study introduces a projector module, a multi-layer perceptron designed to align BEV features across modalities into a shared representation space. The model's robustness is evaluated across 13 sensor corruption scenarios, including cases of missing inputs and corruption in camera, LiDAR, or both modalities simultaneously. The study provides in-depth comparisons with related work, rigorous ablation studies assessing the contributions of the MSM training scheme and projector module, and a t-SNE visualization.

**Strengths:**

1. **Originality:** The study proposes simple design changes like MSM batch augmentation and modality switching which are new to HD map construction literature.

2. **Quality & Significance:**
- Strong empirical results compared to prior work.
- The paper includes ablation experiments on several of the components.
- Extensive robustness analysis with 13 types of sensor corruptions.

**Weaknesses:**

1. **Motivation and Hypothesis:** The necessity for generalizability between C, L and C+L configurations is questionable. Typically, sensor configurations in autonomous systems are determined early, based on cost and performance considerations, making the flexibility to switch between modalities less relevant. While modality switching could theoretically enhance robustness, the current approach requires prior knowledge of the corrupted modality, which can be challenging to detect autonomously in real-world scenarios, especially with complex corruptions like LiDAR crosstalk.
2. **Limited Novelty:**
- The MSM scheme essentially augments each batch by including samples in all three modalities (C/L/C+L), resembling data augmentation rather than a fundamentally novel training strategy. The reported performance improvements could result from this increased data (as the training time increases significantly; table 2) rather than an intrinsic improvement in the methodology.
- The projector module is conceptually similar to BEV encoders, such as those in BEVFusion [1], which also align modality features to address misalignment. While the implementation may differ, the underlying purpose remains the same, raising questions about the novelty of this component in addressing feature misalignment.
- The claim that this is the first study to explore robustness in HD map construction (line 161) overlooks relevant prior work, such as the RoboDrive challenge [2] (and subsequent studies based on this), which addresses similar sensor corruption scenarios and includes additional corruption types.

[1] Liang, T., Xie, H., Yu, K., Xia, Z., Lin, Z., Wang, Y., ... & Tang, Z. (2022). Bevfusion: A simple and robust lidar-camera fusion framework. Advances in Neural Information Processing Systems, 35, 10421-10434.

[2] Kong, L., Xie, S., Hu, H., Niu, Y., Ooi, W. T., Cottereau, B. R., ... & Xu, Y. (2024). The robodrive challenge: Drive anytime anywhere in any condition. arXiv preprint arXiv:2405.08816.

**Questions:**

1. How does the modality switching operate in practice? Is it automated, or does it require prior information about which modality to use, particularly in corruption cases?
2. Was corrupted data used solely for zero-shot analysis, or was it included in training? If included, was the MapTR baseline also trained with this data?
3. Could the authors clarify the total training data used (Table 2) and comment on whether the performance boost might stem from increased data, given that MSM triples the sample count per scenario?
4. How does the projector module differ from the BEV encoder used in BEVFusion?
5. Could the authors comment on whether modality switching is the best approach for handling corruptions, given that even a corrupted sensor may still provide valuable information, and completely discarding its data might not be optimal?

 I am open to updating my rating if the authors provide reasonable responses to these questions and address the mentioned weaknesses.

---

> ### Author Response · Authors · 2024-11-19
> **Thanks and Response to Reviewer dUuw(1/2)**
>
> Thank you for reviewing our paper and for your positive feedback, particularly your comment that our method exemplifies '**_High-Quality Quality and Important Significance_**.' We will incorporate all your suggestions into the revised version of the paper.
>
> >Q1: How does the modality switching operate in practice? Is it automated, or does it require prior information about which modality to use, particularly in corruption cases?
>
> **A1**: Thanks for your comments. First, we do not require prior knowledge of the corrupted modality. All results reported in Figures 5/9 and Tables 12-17 are indeed obtained using the **_Uni-Map Camera-LiDAR Fusion model_**. Moreover, Uni-Map Fusion model shows stronger robustness against the 13 types of camera-LiDAR corruptions we designed, compared to the MapTR Fusion model and HiMap Fusion (detailed results are in Figures 5 and 9). Please note that the modality switching operates automatically and is only triggered when data from one modality is unavailable. Implementing this strategy can enhance performance in such cases by allowing the model to rely on available data from the other modality.
>
> >Q2: Was corrupted data used solely for zero-shot analysis, or was it included in training? If included, was the MapTR baseline also trained with this data?
>
> **A2**: Thanks for your comments. The corrupted data was used solely for zero-shot analysis and was not included in the training process.
>
>
>
> >Q3: Could the authors clarify the total training data used (Table 2) and whether the reported performance improvements may be due to the increased sample count from MSM, rather than intrinsic improvements in the methodology?
>
> **A3**：Thank you for your insightful question. The training data aligns with the nuScenes training dataset. Our all-in-one model utilizes both camera and LiDAR modalities, **_identical to MapTR-F_**. The key difference lies in the decoder’s input features: our model **_reuses_** camera and LiDAR features along with the fused representation, incurring minimal additional computation. This overhead is **_significantly less than the threefold_** increase required for tripling the encoder’s input (**_21h57m vs. 47h12m_**), as shown in the table below. Nevertheless, we trained the MapTR-F model using three times the data volume by increasing the number of epoches, and the experimental results are as follows. Despite this increase in data, our Uni-Map Fusion(mAP:68.1) still outperforms the MapTR-F (mAP:65.4) model. The performance improvement is primarily attributed to MSM concatenates the features along the batch dimension, allowing the stacked BEV features to be input into the same map decoder. This approach enhances the diversity of the BEV feature space, ultimately improving the model’s generalization and robustness across various input configurations. Therefore, merely increasing the amount of data will not lead to the reported performance gains. The novel MSM module and projector module introduced in this paper are crucial for enhancing performance and robustness.
>
> |Method | Data volume | AP_ped. |AP_div. |AP_ped. |mAP |Training Time|
> |  :----:   |    :----:   |  :---: |  :---: |  :---: |  :---: |  :---: |
> | MapTR-F |x1| 55.9|62.3|69.3|62.5|15h44m
> | MapTR-F |x3|61.5|64.2|70.6|65.4|47h12m
> | Uni-Map-F |x1|64.4|66.8|73.2|68.1|21h57m|

---

> > ### Author Response · Authors · 2024-11-19
> > **Thanks and Response to Reviewer dUuw(2/2)**
> >
> > >Q4: How does the projector module differ from the BEV encoder used in BEVFusion?
> >
> > **A4**: Thank you for your feedback. BEVFusion primarily focuses on fusing features from different modalities, with the goal of creating a single fused feature map that effectively utilizes the complementary information between these modalities. In contrast, our method focuses on aligning features from different modalities (camera, LiDAR, and their fusion), aiming to achieve a unified model that can effectively handle various input modality configurations. The technical details are as follows: In the BEVFusion model, camera and LiDAR BEV features are concatenated along the **_feature dimension_** and processed through a convolutional network to **_generate fused features for prediction_**. BEVFusion primarily focuses on creating these fused features. In contrast, our approach passes camera BEV features, LiDAR BEV features, as well as fused BEV features through a projector module to **_align them in the shared feature space_**. After alignment, we concatenate the features along the **batch dimension_**, resulting in a 3B batch that is sent to the prediction head. As outlined in the Implementation section (lines 318-320), during training, ground truth labels are duplicated to match the stacked feature map from Equation 4. In summary, the key difference lies in the BEV encoder design of BEVFusion, which uses convolution to fuse modalities across the channel dimension. In contrast, our projector module is specifically designed to align modalities (camera, LiDAR, and fused BEV features) along the batch dimension, representing a **_fundamentally different approach_**. We appreciate your attention to this distinction.
> >
> >
> > >Q5: Could the authors comment on whether modality switching is the best approach for handling corruptions, given that even a corrupted sensor may still provide valuable information, and completely discarding its data might not be optimal?
> >
> > **A5**: Thank you for your comments. We fully agree that "even a corrupted sensor may still provide valuable information." As stated in Section 4.4, Robustness of Multi-Sensor Corruptions, we evaluate the **_Uni-Map Camera-LiDAR Fusion model (Uni-Map-F) without discarding data from corrupted modalities_**. In fact, all results presented in Figures 5 and 9, as well as Tables 12-17, are generated using the Uni-Map-F model. As a result, our model demonstrates strong robustness against both complete sensor failures and corruptions. It is important to note that the modality switching strategy is **_only activated_** in cases of complete sensor failure (i.e., missing camera or LiDAR). We will clarify this further in the revised version. Thank you again for your valuable feedback!
> >
> >
> > >Q6: About Motivation and Hypothesis.
> >
> > **A6**：Thank you for your comments. We aim to reduce the effort required to deploy models for different vehicle types. Specifically, our method enables the training of a single model that can be deployed across various vehicle types with different sensor configurations (camera, LiDAR, and fusion), eliminating the need to train separate models for each vehicle type. Additionally, we address LiDAR crosstalk directly in the Uni-Map Fusion model without the need to detect corrupted sensors. Importantly, we do not require any prior knowledge of sensor corruption. All results reported in Figures 5 and 9, as well as Tables 12-17, are generated using the Uni-Map Camera-LiDAR Fusion model. The modality switching operates automatically and is triggered when data from one modality is unavailable. We will provide clarification in the revised version of the paper.
> >
> >
> > >Q7: The claim that this is the first study to explore robustness in HD map construction (line 161) overlooks relevant prior work, such as the RoboDrive challenge [2] (and subsequent studies based on this), which addresses similar sensor corruption scenarios and includes additional corruption types.
> >
> > **A7**: The RoboDrive challenge explores five tracks: Track 1 - Robust BEV Detection, Track 2 - Robust Map Segmentation, Track 3 - Robust Occupancy Prediction, Track 4 - Robust Depth Estimation, and Track 5 - Robust multimodal BEV Detection. Tracks 1 to 4 focus exclusively on scenarios involving camera sensor corruption, while Track 5 addresses 3D object detection in a multi-modal context. However, the RoboDrive challenge does not specifically examine the robustness of HD map construction tasks. Additionally, this paper primarily focuses on the multimodal robustness of HD map construction. We appreciate your feedback and will include a discussion of relevant work based on this challenge in the related work section of our revised manuscript.

---

> > > ### Comment · Reviewer_dUuw · 2024-11-22
> > >
> > > Thank you, authors, for addressing my comments. Your responses clarify most of my concerns and provide additional insights into the study. I suggest that, if the manuscript is accepted, you incorporate some of this information in the final camera-ready version. While I remain unconvinced about some aspects of the motivation and hypothesis, the work demonstrates significant merit and contributions to the field. Based on this, I have updated my rating to 6. Congratulations on this great work!

---

> > > > ### Author Response · Authors · 2024-11-26
> > > > **Thanks and Response to Reviewer dUuw**
> > > >
> > > > Thank you very much for your valuable suggestions and positive feedback! We will revise the final version according to these constructive discussions.

---

### Official Review · Reviewer_KwFS · 2024-11-09

**Soundness:** 3
**Presentation:** 4
**Contribution:** 1
**Rating:** 5
**Confidence:** 4

**Summary:**

This paper proposes a unified robust high-definition map construction network (Uni-Map), a single model suitable for all input configurations. They designed a novel Mixed Stack Modal (MSM) training scheme, enabling the map decoder to effectively learn information from camera, LiDAR, and fusion features. This paper also introduced a projection module to align BEV features from different modalities into a shared space, enhancing representation learning and overall model performance. During inference, the model employs a switching modality strategy to ensure compatibility with various modalities. The experiments show the effectiveness.

**Strengths:**

(1)This paper is well-written, and the experiments have also shown effectiveness.
(2)The structure of this paper is clear and intelligible, making it easy to apply in real-world scenarios.

**Weaknesses:**

(1) The method presented in this paper is straightforward and lacks new insights. For example, the projector module is supervised (via gradient back-propagation) to implicitly align BEV features from different modalities and inputting stacked BEV features into the same map decoder increases the diversity of the BEV feature space. These operations are fairly obvious in sensor-fusion based methods. How to verify the MLP-based projector can achieve the alignment? I think the misalignment is mainly caused by the inaccurate Camera-to-BEV View Transform process? Why not apply the projector only on the camera BEV feature?  Additionally, is the stack operation on BEV features just concatenation? How does this differ from the fusion methods like BEVfuison[1] or Pointaugmenting[2]?
(2) About the Inference Phase of the switching modality strategy,  what training strategy (e.g., loss functions) is used to ensure the effectiveness of the switching modality strategy during inference?

[1] Liu Z, Tang H, Amini A, et al. Bevfusion: Multi-task multi-sensor fusion with unified bird's-eye view representation[C]//2023 IEEE international conference on robotics and automation (ICRA). IEEE, 2023: 2774-2781.
[2]Wang C, Ma C, Zhu M, et al. Pointaugmenting: Cross-modal augmentation for 3d object detection[C]//Proceedings of the IEEE/CVF conference on computer vision and pattern recognition. 2021: 11794-11803.

**Questions:**

Please see the Weaknesses section.

---

> ### Author Response · Authors · 2024-11-19
> **Thanks and Response to Reviewer KwFS(1/2)**
>
> Thank you for reviewing our paper and for your positive feedback, particularly your comment that our method exemplifies '**_Effective Experiments and Clear Structure_**'. We will incorporate all your suggestions into the revised version of the paper.
>
> >Q1： The motivation of our paper.
>
> **A1**:Thanks for your comments. We first figure out two critical issues in the current HD map construction task: **_high costs due to separate training and deployment for each input configuration, and low robustness when sensors are missing or corrupted_**. To address these issues, we propose the first high-performance all-in-one solution, i.e., one model for various input configurations (camera-only, LiDAR-only, and camera-LiDAR Fusion). The core components of our method (Uni-Map) include the Mixture Stack Modality (MSM) training scheme and the projector module. Specifically, MSM inputs stacked BEV features into the same map decoder, increasing the diversity of the BEV feature space accessible to the decoder module, thereby improving the generalization ability and robustness of the model in different input configurations (See Tab. 3 and Fig. 5/9). Furthermore, we propose a simple yet effective projector module to align BEV features from different modalities into a shared space (See Fig. 7). As shown in Fig. 7, before the Shared Projector module, the camera BEV and LiDAR/Fused BEV features are clearly separated. After the shared projector module, the BEV features from different modalities are well aligned in a shared/similar space. In addition, we also conducted a thorough ablation study to demonstrate the effectiveness of the Uni-Map model's core modules MSM scheme (See Tab. 3) and Shared Projector module (See Tab. 4). Overall, the Uni-Map model achieves significant performance improvements over SOTA approaches (See Tab. 1) and stronger robustness (See Fig. 5/9), with less training time and fewer parameters for various input configurations (See Tab. 2), while maintaining the same inference speed and memory footprint (See Appendix Tab. 6 and Tab. 7).
>
> >Q2: Are the stack operations on BEV features merely concatenation, and how do they differ from sensor-fusion methods like BEVFusion[1] or PointAugmenting[2]?
>
> **A2**:  Thank you for your feedback. References [1] and [2] primarily focus on fusing features from different modalities, with the goal of creating a single fused feature map that effectively utilizes the complementary information between these modalities. In contrast, our method focuses on aligning features from different modalities (camera, LiDAR, and their fusion), aiming to achieve a unified model that can effectively handle various input modality configurations. The technical details are as follows: In the BEVFusion model, camera and LiDAR BEV features are concatenated along the **_feature dimension_** and processed through a convolutional network to **_generate fused features for prediction_**. BEVFusion primarily focuses on creating these fused features. In contrast, our approach passes camera BEV features, LiDAR BEV features, as well as fused BEV features through a projector module to **_align them in the shared feature space_**. After alignment, we concatenate the features along the **_batch dimension_**, resulting in a 3B batch that is sent to the prediction head. As outlined in the Implementation section (lines 318-320), during training, ground truth labels are duplicated to match the stacked feature map from Equation 4. In summary, the key difference lies in the BEV encoder design of BEVFusion, which uses convolution to fuse modalities across the channel dimension. In contrast, our projector module is specifically designed to align modalities (camera, LiDAR, and fused BEV features) along the batch dimension, representing a **_fundamentally different approach_**. We appreciate your attention to this distinction.

---

> ### Author Response · Authors · 2024-11-19
> **Thanks and Response to Reviewer KwFS(2/2)**
>
> >Q3: How to verify the MLP-based projector can achieve the alignment?
>
> **A3**: Thanks for your comments. First, by stacking BEV features from different modalities that **_share_** the same map decoder and ground truth labels, the projector module is **_supervised_** (via gradient back-propagation) to **_implicitly align_** BEV features from different modalities in the **_shared feature space_**.
> After alignment, we concatenate the features along the batch, allowing the stacked BEV features to be input into the same map decoder, which enhances the diversity of the BEV feature space and improves the model’s generalization and robustness across different input configurations. As can be seen, Fig.7(a) Before Projector module shows that camera BEV (blue) and LiDAR/Fusion BEV (red/green) features are clearly separated, indicating that although in the same space, camera BEV features, LiDAR BEV features, and fused BEV features can still be misaligned to some extent due to the inaccurate depth in the view transformer and the large modality gap. Fig.7 (b) After the projector module, the BEV features from different modalities are well aligned in a shared/similar space, i.e., red (Fused BEV), blue (Camera BEV), and green (LiDAR) circles are close together after the projector module. Moreover, Tab.3 shows that we can significantly improve the model performance by training the model with the projector module using the RSM or MSM training scheme. Therefore, it is demonstrated that the MLP-based projector work well and can align BEV features from different modalities into a shared space, thereby enhancing representation learning.
>
>
> >Q4: I think the misalignment is mainly caused by the inaccurate Camera-to-BEV View Transform process? Why not apply the projector only on the camera BEV feature?
>
> **A4**: Thanks for your comments. The misalignment arises from both spatial mismatches due to **_inaccuracies in camera-to-BEV transformations_** and sensor calibration, as well as a **_substantial modality gap_** between camera images and LiDAR point clouds. Consequently, projecting camera BEV features into a fixed LiDAR space, by solely applying the projector exclusively to the camera BEV features, is **_challenging to achieve effective alignment_**. This paper introduces a projector module to align BEV features from different modalities into a shared space, which is crucial for enhancing the overall performance of our model.
>
> >Q5: About the Inference Phase of the switching modality strategy, what training strategy (e.g., loss functions) is used to ensure the effectiveness of the switching modality strategy during inference?
>
> **A5**：Thanks for your comments. During training, we employ the MSM strategy, which effectively ensures the performance of the switching modality during inference without the need for additional strategies. This is primarily because MSM concatenates the features along the batch dimension, allowing the stacked BEV features to be input into the same map decoder. This approach enhances the **_diversity_** of the BEV feature space, ultimately improving the model’s generalization and robustness across various input configurations. As detailed in the Implementation section (lines 318-320), during training, ground truth labels are **_duplicated_** and stacked to create a 3B batch dimension, aligning with the stacked feature map from Equation 4. Thank you for your insightful question.

---

> > ### Comment · Reviewer_KwFS · 2024-11-22
> >
> > Thanks for your responses, I have two other concerns:
> > (1)For Q3: Why did you replace the conv2D operations in BEV used in BEVfusioN/pointaugmenting with the MLP network? What are the advantages of MLP over conv2D for alignment?
> > (2)For Q5:Have you used masking strategies in training, such as randomly masking one modality in training? If one modality is unavailable, do you need to reload the parameters in the map decoder?

---

> ### Author Response · Authors · 2024-11-22
> **Thanks and Response to Reviewer KwFS**
>
> Thank you for your insightful follow-up comments.
>
> >Q1: Why did you replace the conv2D operations in BEV used in BEVfusioN/pointaugmenting with the MLP network? What are the advantages of MLP over conv2D for alignment?
>
> **A1**: The replacement of Conv2D operations with an MLP is primarily for effective feature transformation. MLPs, similar to the feedforward neural networks in transformers, excel at learning complex, non-linear mappings while maintaining the original input structure, thus preserving spatial context. They also offer a global perspective, are easier to implement, and adapt well to irregular data, making them more suitable for alignment tasks in multi-modal scenarios.
>
> >Q2: Have you used masking strategies in training, such as randomly masking one modality in training? If one modality is unavailable, do you need to reload the parameters in the map decoder?
>
> **A2**: To systematically evaluate the effectiveness of the MSM training scheme, we train the model using different schemes and report the mAP results in Tab. 3. In addition to MSM, we introduced the Random Select Modality (RSM) training scheme, which randomly selects inputs from one of the BEV feature maps—either Camera BEV features, LiDAR BEV features, or Fused BEV features—effectively acting as a masking strategy. The results of the RSM training scheme are inferior to the MSM training scheme under both settings (with and without the Projector). This demonstrates the MSM training scheme's advantage in enhancing the map decoder's effective use of camera, LiDAR, and fused features. This increases the diversity of the BEV feature space, resulting in a high-performance integrated model. Moreover, if one modality is unavailable, we will perform inference using the BEV feature from the other modality, employing the switching modality strategy described in formula (6). **_No parameter reloading is needed since we have only one model for various input configurations._**
>
> We will revise the final version according to these constructive discussions. Thanks again and feel free to inform us for any further questions and discussions.

---

> > ### Author Response · Authors · 2024-11-27
> > **Looking forward to further feedback**
> >
> > Dear Reviewer KwFS,
> >
> > Thank you again for your valuable comments and suggestions, which are very helpful to us. We have responded to the proposed concerns. We hope that it has helped to address the concerns you have raised in your review.
> >
> > We understand that this is quite a busy period, so we sincerely appreciate it if you could take some time to return further feedback on whether our responses resolve your concerns. If you have any further questions, we are more than happy to address them before the conclusion of the rebuttal phase.
> >
> > Best,
> >
> > The Authors

---

### Comment · Area_Chair_WxXW · 2024-11-27
**Reminder: Last day for author feedback**

This is a reminder that today is the last day allotted for author feedback. If there are any more last minute comments, please send them by today.

---

### Meta-Review · Area_Chair_WxXW · 2024-12-19

**Metareview:**

The authors propose a method for high-definition map construction. The main contribution lies in a Mixed Stack Modal training scheme that allows for different configuration including camera, lidar, or both modalities. The authors employ a projector (MLP) to align modalities into a common latent space. We have read the referee reports, the author responses as well as the manuscript, since there were several points  raised by the referees and the score changes to cause a borderline decision during discussion. The concerns included the motivation, incremental and heavy engineering nature of the work and concerns about the use of the projector (e.g., MLPs) raised by frKwFS, dUuw, and js2w. The proposed method is built on MapTR and the projector bares similarities to existing work in the field, such as BEVFusion and Pointaugmenting [A, B], as well as those used by representation learning such as BYOL [C]. The critical point raised lies in the projector module proposed, which the referees have discussed and concluded that there may be algorithmic concerns and the manuscript lacks analysis for it. We note that while scTZ did show support for the manuscript, stating ``it is a simple and effective approach [and] does not necessarily imply a lack of contribution'', scTZ also agreed with the other referees on the previous points. We would like to clarify that frKwFS, dUuw, and js2w were not claiming a lack of contribution due to simplicity but due to algorithmic soundness, which left the three referees unconvinced of the approach. Nonetheless, we believe the discussion and additional experiments did help to clarify many points that we also were unclear about when reading the paper. We encourage the authors to incorporate the feedback into the next revision.

[A] Liu et al. Bevfusion: Multi-task multi-sensor fusion with unified bird's-eye view representation. ICRA 2023.

[B] Wang et al. Pointaugmenting: Cross-modal augmentation for 3d object detection. CVPR 2021.

[C] Grill et al. Bootstrap Your Own Latent A New Approach to Self-Supervised Learning. NeurIPS 2020.

**Additional Comments On Reviewer Discussion:**

We have read the referee reports, the author responses as well as the manuscript, since there was a vague and short report provided by scTZ who recommended accept against several critical points raised by other referees. The discussion also led to score downgrades to cause a borderline decision. The concerns included the motivation, incremental and heavy engineering nature of the work, and concerns about the use of the projector (e.g., MLPs) raised by frKwFS, dUuw, and js2w. The proposed method is built on MapTR and the projector bares similarities to existing work in the field, such as BEVFusion and Pointaugmenting [A, B], as well as those used by representation learning such as BYOL [C]. The critical point raised lies in the projector module proposed, which the referees have discussed and concluded that there may be algorithmic concerns and the manuscript lacks analysis for it. We note that while scTZ did show support for the manuscript, stating ``it is a simple and effective approach [and] does not necessarily imply a lack of contribution'', scTZ also agreed with the other referees on the previous points. We would like to clarify that frKwFS, dUuw, and js2w were not claiming a lack of contribution for the simplicity but due to algorithmic soundness, which left the three referees unconvinced of the approach.

---

### Decision · Program_Chairs · 2025-01-22

Reject